# Arabidopsis and maize terminator strength is determined by GC content, polyadenylation motifs and cleavage probability

Sayeh Gorjifard [1], Tobias Jores [1], Jackson Tonnies [1,2], Nicholas A. Mueth[1], Kerry Bubb[1], Travis Wrightsman[3], Edward S. Buckler [3,4,5], Stanley Fields [1,6], Josh T. Cuperus [1] & Christine Queitsch [1] ✉

The 3′ end of a gene, often called a terminator, modulates mRNA stability, localization, translation, and polyadenylation. Here, we adapted Plant STARR-seq, a massively parallel reporter assay, to measure the activity of over 50,000 terminators from the plants *Arabidopsis thaliana* and *Zea mays*. We characterize thousands of plant terminators, including many that outperform bacterial terminators commonly used in plants. Terminator activity is species-specific, differing in tobacco leaf and maize protoplast assays. While recapitulating known biology, our results reveal the relative contributions of polyadenylation motifs to terminator strength. We built a computational model to predict terminator strength and used it to conduct in silico evolution that generated optimized synthetic terminators. Additionally, we discover alternative polyadenylation sites across tens of thousands of terminators; however, the strongest terminators tend to have a dominant cleavage site. Our results establish features of plant terminator function and identify strong naturally occurring and synthetic terminators.

A critical challenge is producing enough food for a growing world population that is likely to reach over 9 billion people by the year 2050[1]. Food security can be bolstered through crop engineering, which can improve yields and increase tolerance to pathogens and environmental stresses[2,3]. However, for crop engineering to be successful, precise control over transgene expression is required. In plants, past efforts to optimize critical *cis*-elements have mostly focused on the identification, characterization, and manipulation of upstream regulatory elements such as promoters[4–8]. Although these upstream elements can produce large effects on gene expression, other sequence elements also contribute. For example, the 3′ end of a gene contains sequence motifs necessary for mRNA 3′ end maturation, cleavage, and polyadenylation[9–16]. In keeping with established nomenclature[12,17,18], throughout this manuscript, we refer to sequences

surrounding a cleavage and polyadenylation site as terminators. Terminators also affect mRNA stability, nuclear export, and translation[12]. Moreover, terminators play a central role in transgene silencing mediated by small RNAs[19–23].

Messenger RNA cleavage and polyadenylation are tightly linked to and initiate transcription termination[24–27]; however, termination occurs up to 1 kb downstream of the mRNA cleavage site[28]. The current model of how mRNA 3′ end processing leads to transcription termination combines features of two previously proposed models[29–32]: After mRNA cleavage, conformational changes of RNA polymerase II allosterically reduce its elongation rate (allosteric model). At the same time, the cleavage site enables a 5′ to 3′ exonuclease (Rat1 in yeast, XRN2 in humans, and XRN3 in *Arabidopsis*) to degrade the downstream RNA and initiate transcription termination when it catches up

[1]Department of Genome Sciences, University of Washington, Seattle, WA 98195, USA. [2]Graduate Program in Biology, University of Washington, Seattle, WA 98195, USA. [3]Section of Plant Breeding and Genetics, Cornell University, Ithaca, NY 14853, USA. [4]Agricultural Research Service, United States Department of Agriculture, Ithaca, NY 14853, USA. [5]Institute for Genomic Diversity, Cornell University, Ithaca, NY 14853, USA. [6]Department of Medicine, University of Washington, Seattle, WA 98195, USA. ✉e-mail: queitsch@uw.edu

with the RNA polymerase (torpedo model). The exact site of transcription termination can further be affected by RNA polymerase II pause sites, R loops formed between DNA and the nascent RNA, and DNA-binding proteins[31,33].

Although the choice of terminator significantly impacts transgene expression, only a few terminators − often those derived from viruses or bacteria − are commonly used in crop engineering. These viral or bacterial sequences make transgene silencing more likely[22,34]. Compared to the commonly used *Agrobacterium tumefaciens* nopaline synthase terminator (tNOS), the terminator from the *Arabidopsis thaliana* heat shock protein 18.2 results in increased transgene expression and mRNA stability, and decreased silencing[23,35–38]. This finding highlights the need to characterize and optimize native plant terminators as building blocks for crop engineering applications[18,22,39].

Plant terminators have been characterized by analysis of wild-type and mutated versions of a handful of terminators, from both plant and non-plant sources, or by searching for motifs enriched in genomic sequences surrounding mRNA cleavage sites. These studies established three main *cis*-acting elements required for 3′ end processing[10,12,13,40,41]. The first element is the far upstream element (FUE), which resides in a U-rich region located 30–150 nucleotides upstream of the cleavage site and often contains one or more UGUA sequences. The second is the near upstream element (NUE), which resides in an A-rich region located 10–30 nucleotides upstream of the cleavage site and contains the polyadenylation signal AAUAAA. The third is the cleavage element (CE), which contains the cleavage site, formed by a UA or CA dinucleotide embedded in a U-rich region. Although the general pattern and nucleotide preferences of *cis*-acting elements in plant terminators are known, their relative contributions to terminator strength and species specificity have not been determined.

Here, we characterized over 50,000 terminators from the model plant *Arabidopsis thaliana* and the crop plant maize (*Zea mays*). We identified sequence features contributing to terminator strength and quantified their effect on transcript levels. By measuring terminator strength in two assay systems—tobacco leaves and maize protoplasts—we detected similarities and differences in the terminator grammar of monocotyledonous and dicotyledonous plants. Leveraging our large dataset, we trained computational models that accurately predict terminator strength. We used these models to design robust synthetic plant terminators for future crop engineering efforts.

## Results

### Measuring the strength of plant terminators with Plant STARR-seq

To assess the strength of plant terminators at high throughput, we used Plant STARR-seq, a massively parallel reporter assay that measures the activity of cis-regulatory elements[4,5]. The average 3′ untranslated region (UTR) length in *Arabidopsis* and maize is 242 bp and 310 bp, respectively (Srivastava 2018; Jafar 2019). However, due to technical limitations in DNA array-synthesis, we were limited to sequences of 170 nucleotides. Previous studies on plant terminators revealed that most elements required for efficient mRNA 3′ end processing reside within approximately 150 bp upstream of the cleavage and polyadenylation site[10,12,13,16,40,41]. Furthermore, studies in yeast and animals revealed that sequence elements downstream of the cleavage site can also affect polyadenylation[12,14,41]. Although such elements have not been reported in plants, we included the 20 nucleotides downstream of the cleavage site in our candidate sequences to test if they have an effect of terminator activity. For these reasons, we defined a terminator as the 170 nucleotide sequence from position −150 to +20 relative to a cleavage and polyadenylation site (position 0) in this study. We used experimentally derived primary cleavage sites to select terminator sequences from wild-type Arabidopsis and maize[42–44]. For the 3754 Arabidopsis genes that were not represented in these

datasets, we used the end of the 3′ untranslated region (UTR) annotation in TAIR10 as the cleavage site. Since alternative polyadenylation plays an important role in gene regulation by creating different mRNA isoforms[11], we sought to investigate the strength of both primary and secondary polyadenylation sites. Therefore, we included experimentally-derived, secondary cleavage sites supported by at least 30% of the total reads of a gene[42–44]. As expected, the 24,529 Arabidopsis terminators and the 30,092 maize terminators displayed the distinctive nucleotide composition preferences of terminator sequences[41] and predominantly resided in the annotated 3′ UTR region (Supplementary Fig. 1a–d).

The library of plant terminators was array-synthesized and cloned downstream of the coding sequence of a barcoded green fluorescent protein (GFP) reporter gene driven by the cauliflower mosaic virus 35S promoter (Fig. 1a). The plasmid library was transiently expressed in tobacco leaves or maize protoplasts. After 1–2 days of incubation, the reporter mRNA was extracted, and next-generation sequencing was used to count the reporter barcodes in the input DNA and the extracted RNA. Strong terminators yield higher transcript levels due to improved 3′ end processing or transcript stability as compared to weak terminators[12,41]. The number of transcripts (i.e. RNA reads) per DNA template (i.e. DNA reads) is therefore a direct measure of the strength of a terminator. We define terminator strength as the enrichment of barcodes in RNA over DNA normalized to the enrichment of a control construct containing the 35S terminator. Thus, terminator strength in our assay reflects both transcriptional activity and RNA stability, and tends to have low values due to the normalization to the highly active 35S terminator.

We performed two biological replicates in tobacco leaves and maize protoplasts. The results were highly correlated in both assay systems (Fig. 1b, c). Similarly, retesting of over 400 terminators in a second, independent library showed that the replicates in this validation experiment were highly correlated with each other and with the results of the first, large-scale experiment (Supplementary Fig. 2). Therefore, we used the average terminator strength from both replicates of the large-scale experiment for all further analyses. Terminator strength spanned a wide range of activity, allowing us to disentangle the signals that contribute to this strength. In tobacco leaves, we observed more than 64-fold difference between strong and weak terminators. Similar to previous studies[4,5], the dynamic range was lower in maize protoplasts, in which we detected a 16-fold difference between strong and weak terminators.

To ensure that we measured terminator strength, we included negative controls derived from coding regions in *Arabidopsis* and maize and from randomized sequences. The random sequences were generated such that their overall (global random) or per-position (positional random) nucleotide frequencies resembled that of an average *Arabidopsis* or maize terminator (Supplementary Fig. 1e-g). In both tobacco leaves and maize protoplasts, sequences derived from plant terminators outperformed these coding sequences and random controls (Fig. 1d, e). Of all control sequences, the positional random sequences showed the greatest strength, likely because they most closely resemble actual terminators. Together, these findings demonstrate that our assay captures bona fide terminator strength.

We sought to confirm our measure of terminator strength by Plant STARR-seq with an orthogonal assay. To do so, we conducted a dual-luciferase assay in both tobacco leaves and maize protoplasts for several weak, intermediate, and strong terminators. We observed a strong correlation between terminator strength measured by the two assays (Supplementary Fig. 3b, d). Strong terminators yielded higher nanoluciferase activity than weak terminators (Supplementary Fig. 3a, c).

Most plant transgenes use viral or bacterial terminators, such as the 35S, Ag7, NOS, and MAS terminators[45]. We included these terminators in our library in their full form (not limited to 170 nucleotides) as

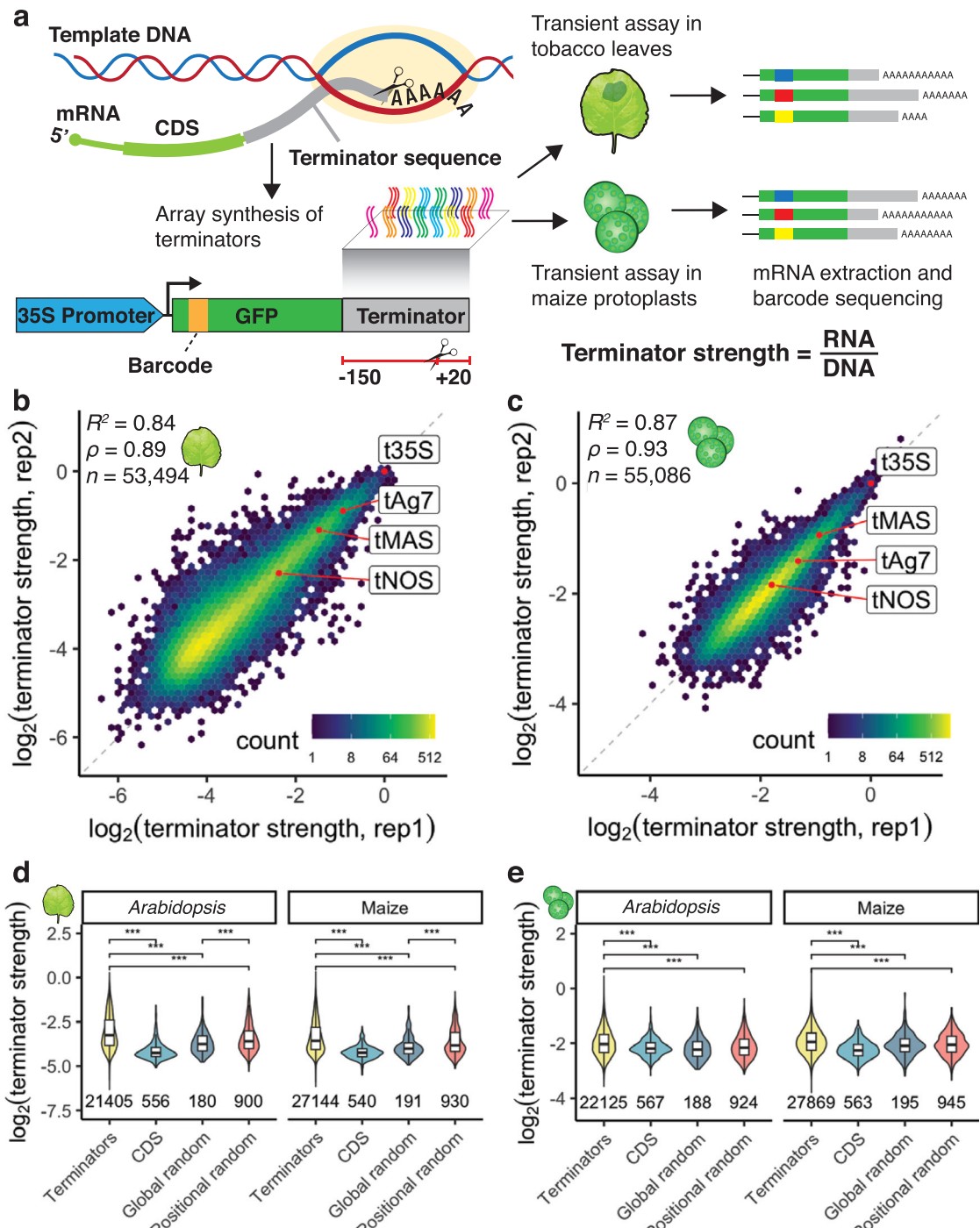

**Fig. 1 | Plant STARR-seq measures terminator strength in tobacco leaves and maize protoplasts. a** Terminator sequences (bases −150 to +20 relative to the cleavage and polyadenylation site) were array-synthesized and cloned downstream of a barcoded GFP reporter gene driven by the 35S promoter. After transient expression of the plasmid library in tobacco leaves or maize protoplasts, mRNA was extracted for barcode sequencing. We define terminator strength as the enrichment of barcodes in the extracted mRNA over the input DNA normalized to the strength of the 35S terminator. **b**, **c** Hexbin plots (color represents the count of points in each hexagon) of the correlation between two biological replicates of Plant STARR-seq in tobacco leaves (**b**) or maize protoplasts (**c**). Commonly used terminators are highlighted in red. Pearson's $R^2$, spearman's $\rho$, and number ($n$) of terminators are indicated. **d**, **e** Violin plots, box plots, and significance levels of

terminator strength in tobacco leaves (**d**) or maize protoplasts (**e**) for plant terminators (Terminators) compared to sequences from coding regions (CDS) and randomized sequences with an overall (Global random) or per-position (Positional random) nucleotide frequency similar to an average *Arabidopsis* or maize terminator. Violin plots represent the kernel density distribution and the box plots inside represent the median (center line), upper and lower quartiles and 1.5× the interquartile range (whiskers) for all corresponding terminators. Numbers at the bottom of each violin indicate the number of terminators in each group. Significant differences between two samples were determined by the two-sided Wilcoxon rank-sum test and are indicated: *$p \leq 0.05$, **$p \leq 0.01$, ***$p \leq 0.001$, NS, not significant. Exact $p$ values are listed in Supplementary Data 3.

a reference for the activity of the plant terminators. The 35S terminator, the only viral sequence in our library, outperformed almost all plant terminators. However, some plant terminators (8 in tobacco leaves and 20 in maize protoplasts) exceeded the activity of the 35S terminator. The bacterial Ag7, NOS, and MAS terminators were far weaker than the viral 35S terminator. In tobacco leaves, we found 2,224 plant terminators that were stronger than the MAS terminator and 9389 plant terminators that were stronger than the NOS terminator. In maize protoplasts, we found 1,369 plant terminators that were stronger than the MAS terminator and 19,585 terminators that were stronger than the NOS terminator. The strongest plant terminators are attractive alternatives to the bacterial terminators commonly used in plant transgenes.

## Terminator strength correlates not with gene expression but with gene function

Next, we asked if our data could give insights into the importance of terminator strength for gene expression. We compared terminator strength determined by Plant STARR-seq with metrics related to mRNA levels, stability and degradation. We found only weak correlation between terminator strength and gene expression[46], mRNA half-life[47], or nascent transcription[48] (Supplementary Fig. 4). These findings corroborate reports that upstream *cis*-regulatory elements, like enhancers, promoters, and the 5′ UTR, drive large-scale gene expression changes while the 3′ UTR and terminators fine-tune expression levels[49,50]. However, the choice of terminator can have a significant impact on the expression of transgenes[23,51].

While the correlation between terminator strength and gene expression was low, we wondered whether the function of a gene and the strength of its terminator were more correlated. To address this question, we performed gene ontology (GO) term enrichment analysis on the genes associated with the strongest 10% of terminators from *Arabidopsis* or maize. For both species, we found a significant enrichment (adjusted *p*-value < 0.05) for genes related to metabolism and response to stimulus (Supplementary Fig. 5a–d). Terminators derived from oxidoreductase-related and stress-responsive genes in *Arabidopsis* were overrepresented in the top 10% of terminators ranked by strength in tobacco leaves and maize protoplasts. Maize terminators derived from genes involved in small molecule metabolic processes were overrepresented in the strongest 10% of terminators in both systems. However, we found many more significant GO terms associated with metabolism for the strongest maize terminators in maize protoplasts than in tobacco leaves (Supplementary Fig. 5e, f). The latter finding pointed to possible species-specific differences in terminator strength, prompting us to investigate this possibility further.

## Plant terminator strength is species-specific

mRNA 3′ end processing may differ between monocotyledonous and dicotyledonous plants, as suggested from studies in the 1980s. For example, the gene encoding the small subunit of ribulose 1,5-bisphosphate carboxylase (*rbcS*) from the monocot wheat is improperly polyadenylated when tested in the dicot tobacco[52]. In contrast, the mRNA of the *rbcS* gene from the dicot pea is efficiently polyadenylated in tobacco[53]. Consistent with these earlier findings, we observed a relatively weak correlation ($R^2 = 0.33$) between the strength of a given terminator in the dicot model tobacco leaves versus the monocot model maize protoplasts (Fig. 2a). In tobacco leaves, terminators from the dicot *Arabidopsis* performed significantly better than terminators from the monocot maize. Similarly, in maize protoplasts, maize terminator sequences significantly outperformed *Arabidopsis* terminators (Fig. 2b, c). These observations are consistent with species-specific differences in plant terminator strength.

Based on these observations, we wanted to identify terminators with strong species-specific activity. Since our assay systems show different dynamic ranges, we normalized terminator strength within

each assay system to a scale from 0 (weakest) to 1 (strongest). We defined a variable Ψ as the difference between the normalized terminator strength in tobacco leaves and in maize protoplasts, such that a high Ψ value denotes a terminator that was considerably stronger in tobacco leaves than in maize protoplasts. We found that *Arabidopsis* terminators show a higher average Ψ value than maize terminators (Fig. 2d). We selected the top 10% of *Arabidopsis* terminators ($n = 1,923$) with highest Ψ values, *i.e.* tobacco-specific strength, and the top 10% of maize terminators ($n = 2,439$) with the lowest Ψ values, *i.e.* maize-specific strength, for GO term enrichment analysis. *Arabidopsis* terminators with highly tobacco-specific strength tended to be derived from stimulus- and stress-responsive genes, while maize terminators with highly maize-specific strength tended to be derived from metabolic genes (Fig. 2e).

To understand how species-specific terminator strength is mediated, we used STREME[54] to search for RNA motifs that were enriched in the tobacco-specific *Arabidopsis* terminators (high Ψ) relative to the maize-specific maize terminators (low Ψ), and vice versa. The most enriched motifs in the tobacco-specific *Arabidopsis* terminators were dominated by long stretches of U nucleotides, while the most enriched motifs found in maize-specific maize terminators favored G and C nucleotides (Fig. 2f, g). These findings motivated us to further analyze the effect of nucleotide composition on terminator strength.

## Nucleotide composition affects terminator strength in a position-specific manner

Dicot genomes have a lower GC content than monocot genomes[55]. This bias also holds true for *Arabidopsis* and maize terminator sequences (Fig. 3a). Assessing the impact of GC content on terminator strength, we found that terminators with either high or low GC content were weaker than those with intermediate GC content in both tobacco leaves and maize protoplasts (Fig. 3b, c). There are, however, subtle differences between the two assay systems. In tobacco leaves, terminators with a GC content around 30–35% were the strongest, while in maize protoplasts, terminators with a GC content of approximately 35-40% were the strongest. To substantiate these subtle differences, we measured the terminator strength of random sequences with a GC content of 30%, 40%, 50%, 60%, or 70%. In tobacco leaves, the random sequences with a GC content of 30% performed best, while in maize protoplasts, the random sequences with a GC content of 40% showed the highest strength (Supplementary Fig. 6). These values coincide with the average GC content of *Arabidopsis* terminators (32.5%; the average GC content of tobacco terminators is likely in a similar range) and maize terminators (40.9%). Thus, we conclude that the cleavage and polyadenylation machinery is likely attuned to the species-specific GC content of plant terminators.

Since terminator strength is governed at the RNA level, G and C or A and U nucleotides are not necessarily represented equally. Therefore, we teased apart the positional effect on terminator strength for each of the four nucleotides (Fig. 3d, e). We found that C nucleotides have little to no effect on terminator strength in tobacco leaves or maize protoplasts. High G content was beneficial at the 5′ end of our tested sequences but not tolerated near the cleavage site, especially in tobacco leaves. Conversely, high U content near the 5′ end was associated with lower terminator strength. The system preferences diverged, however, for sequences with high U content starting from 100 nucleotides upstream from the cleavage site: U-rich sequences in this region were associated with increased terminator strength in tobacco leaves but not in maize protoplasts. High A content around 20 nucleotides upstream of the cleavage site was correlated with terminator strength in both systems, probably due to the canonical polyadenylation signal — a short sequence often represented by the hexamer AAUAAA — typically found at this location.

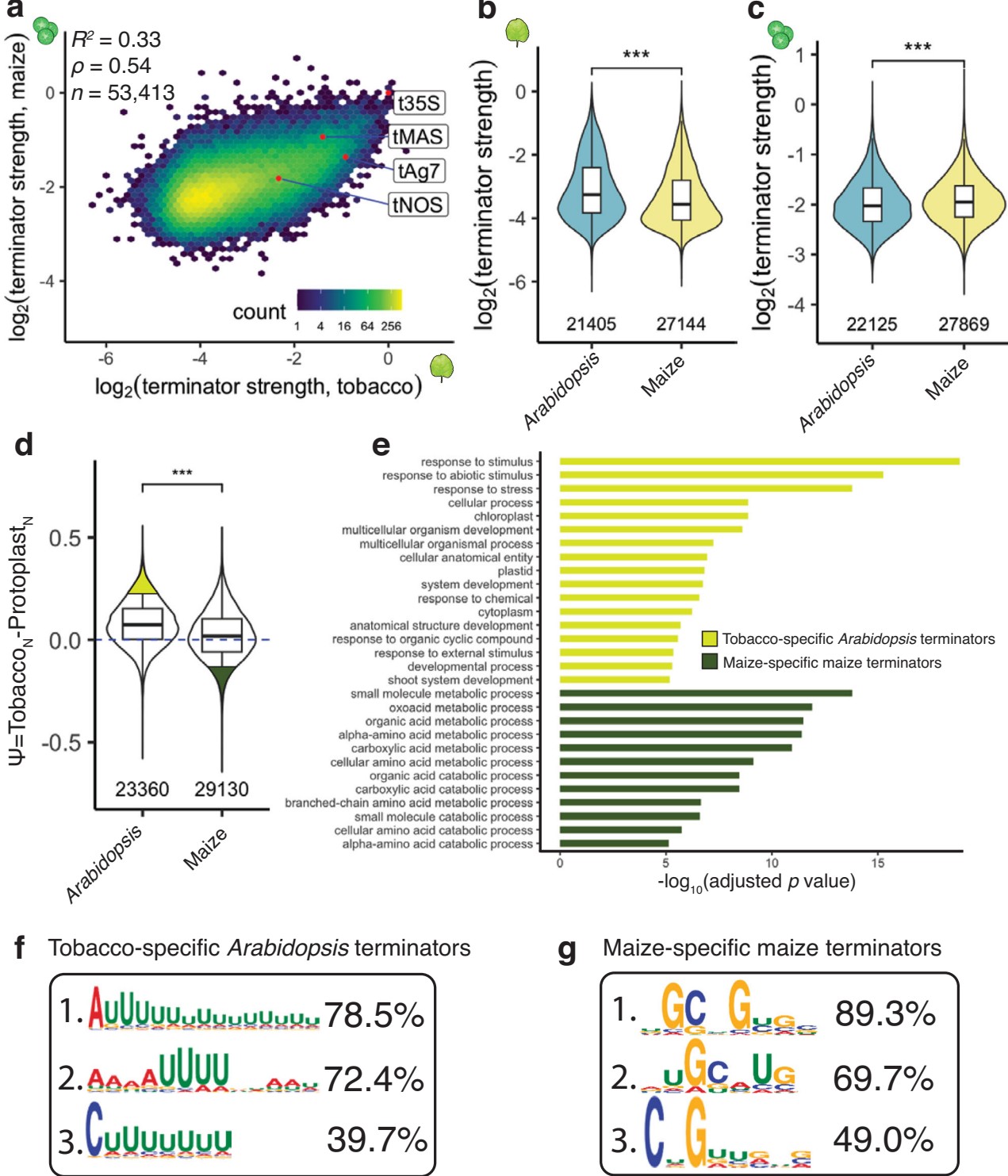

**Fig. 2 | Plant terminator strength is species-specific. a** Hexbin plot comparing terminator strength in tobacco leaves and maize protoplasts. **b, c** Violin plots of the strength of terminators derived from the indicated species in tobacco leaves (**b**) or maize protoplasts (**c**). **d** Violin plot of Ψ, the difference between normalized (0, weakest; 1, strongest) $\log_2$(terminator strength) in tobacco leaves (Tobacco$_N$) and maize protoplasts (Protoplast$_N$). Terminators are grouped by their species of origin. The top 10% of *Arabidopsis* terminators with highest Ψ values (tobacco-specific terminators) and the top 10% of maize terminator with the lowest Ψ values (maize-specific terminators) are highlighted in yellow and green, respectively. **e** GO terms enriched in genes associated with the tobacco-specific *Arabidopsis* terminators or the maize-specific maize terminators highlighted in d. Only GO terms with adjusted *p*-value ≤ 0.0001 are shown. The *p* values were determined using the gprofiler2 library in R with gSCS correction for multiple testing. All enriched GO terms and exact *p* values are listed in Supplementary Data 4. **f, g** Top three motifs enriched in tobacco-specific Arabidopsis terminators relative to maize-specific maize terminators (**f**) and vice versa (**g**), with percentages denoting how many terminators per group had this motif. The hexbin plot in (**a**) and the violin plots, box plots, and significance levels in (**b**–**d**) are as defined in Fig. 1.

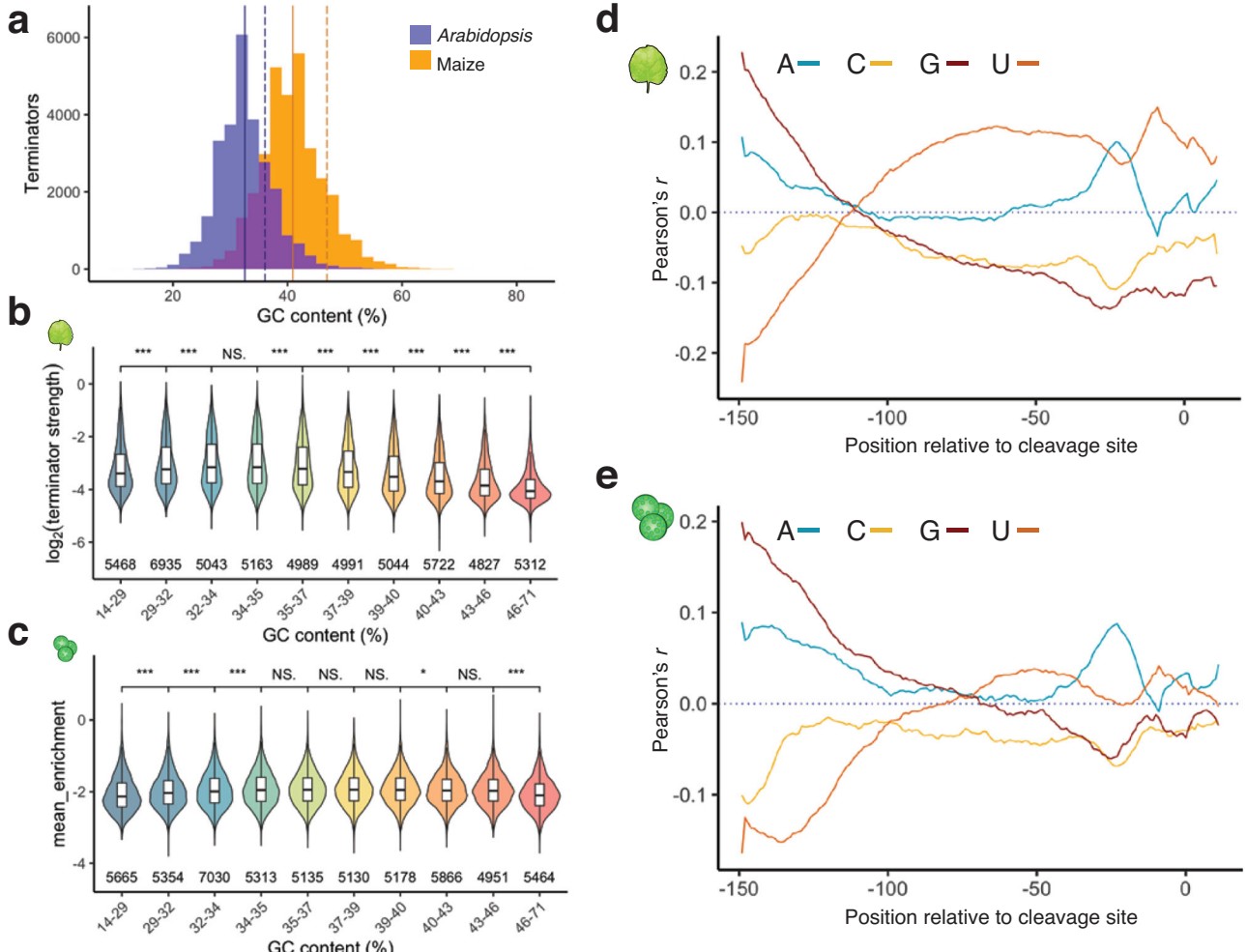

**Fig. 3 | Nucleotide composition affects terminator strength in a species- and position-specific manner. a** Histogram of plant terminator GC content. The solid line indicates the mean GC content of terminators (*Arabidopsis* = 32.53%, maize = 40.89%) and the dashed line indicates the GC content of the genome (*Arabidopsis* = 36.06%, maize = 46.86%). **b, c** Violin plots and box plots (as defined in Fig. 1) of terminator strength in tobacco leaves (**b**) or maize protoplasts (**c**). Terminators were binned by GC content to yield groups of approximately the same size. **d, e** Correlation (Pearson's *r*) between terminator strength in tobacco leaves (**d**) or maize protoplasts (**e**) and the A, C, G, or U content of a ten-base window starting at the indicated position in the plant terminators.

## Canonical cleavage and polyadenylation motifs contribute to terminator strength

Since the strength of a terminator is only partially explained by its nucleotide composition, we searched for sequence motifs that contribute to terminator strength. To this end, we conducted a motif enrichment analysis on the strongest 25% of terminators in both assay systems. The most significantly enriched motifs match canonical cleavage and polyadenylation signals: the UGUA sequence motif from the Far Upstream Element (FUE) and the AAUAAA polyadenylation signal (Fig. 4a). While these motifs are known to affect cleavage and polyadenylation, our assay allowed us to determine the quantitative contribution of each motif to terminator strength.

The UGUA motif is highly prevalent in our library, with 60% to 80% (small differences between the UGUA-containing motifs identified by STREME lead to different numbers of positive terminators) of all terminators containing this motif. The UGUA motif is predominantly located 30 to 40 nucleotides upstream of the cleavage site in both *Arabidopsis* and maize terminators (Supplementary Fig. 7a, b). In tobacco leaves and maize protoplasts respectively, terminators with a UGUA motif were on average 50% and 20% stronger than those without this motif (Fig. 4b, c). Consistent with prior reports[15], the number of UGUA motifs also correlates with terminator strength and cleavage probability (Supplementary Fig. 8e, f).

The polyadenylation signal is less prevalent in our library (15% to 35% of all terminators) but shows a distinct localization profile, with a sharp peak about 20 nucleotides upstream of the cleavage site (Supplementary Fig. 7c, d). The polyadenylation signal contributes significantly to terminator strength. In tobacco leaves and maize protoplasts respectively, terminators with a polyadenylation motif were 40% and 20% stronger than those without this motif (Fig. 4b, c). Since fewer than half of the terminators in our library contain a canonical polyadenylation signal, we asked if similar sequences are functionally equivalent. Terminators with a perfect AAUAAA sequence were stronger than terminators with a sequence that differs by a single nucleotide (carrying a single-nucleotide variant of this motif), although AAUAAG and AAUAAU motifs performed nearly as well. Terminators without any AAUAAA-like sequence were considerably weaker (Supplementary Fig. 8a, b).

In addition to the AAUAAA and UGUA motifs[10,12], we found U- and G-rich motifs enriched in the strongest terminators (Fig. 4a). These U/G-rich motifs are broadly distributed upstream of the polyadenylation signal and at slightly higher frequency just upstream or downstream of the cleavage site (Supplementary Fig. 7e, f). While the U/G-rich motifs share a similar localization pattern, there are striking differences between the motifs discovered in the tobacco leaf and the maize protoplast system. The motif detected in the tobacco leaf data

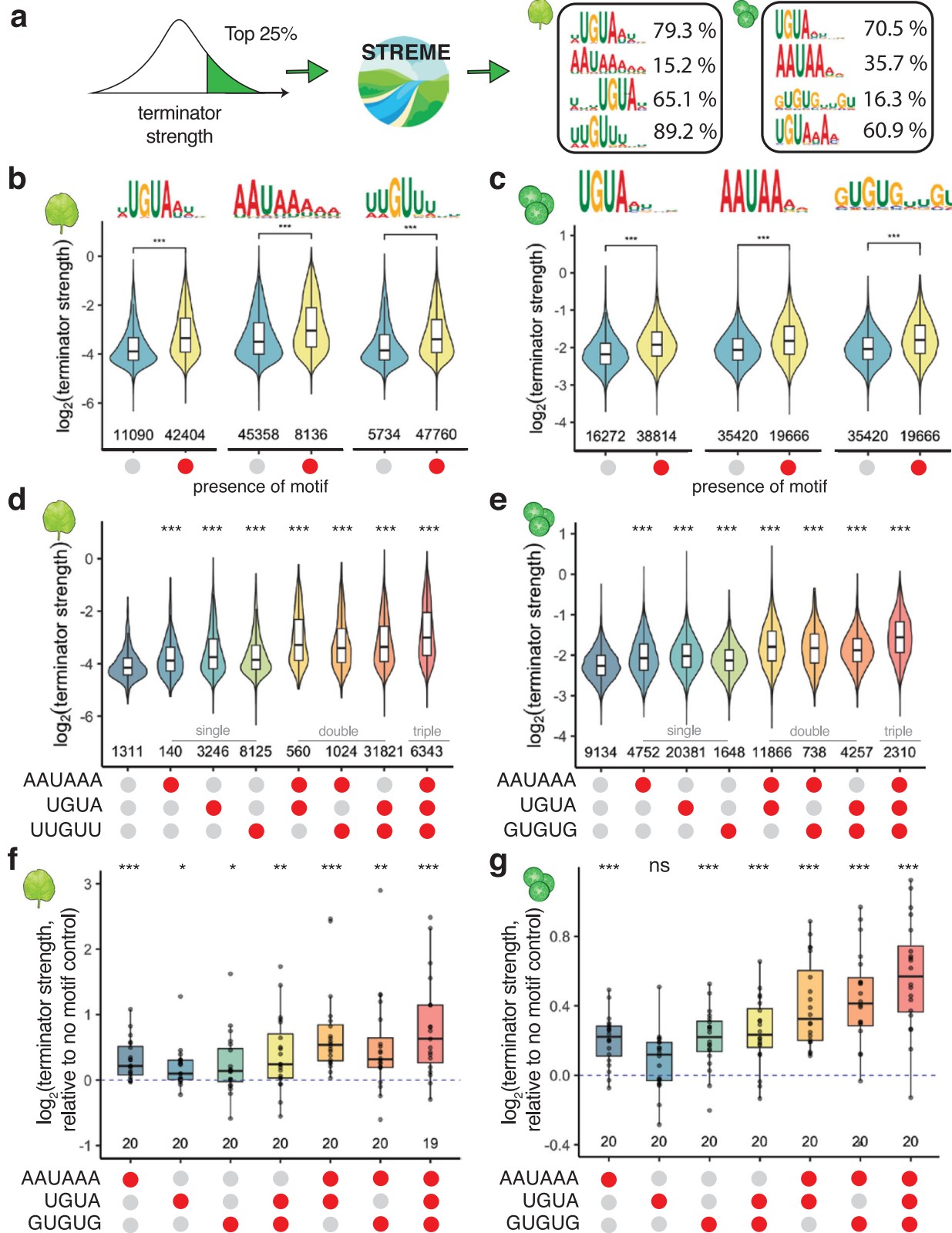

**Fig. 4 | Cleavage and polyadenylation motifs increase terminator strength.**
**a** Sequence logo plots of the top four motifs enriched in the strongest terminators in tobacco leaves (left) or maize protoplasts (right). For each motif, the percentage of *Arabidopsis* and maize terminators harboring it is indicated. **b**–**e** Violin plots of terminator strength in tobacco leaves (**b**, **d**) or maize protoplasts (**c**, **e**) with (red circle) or without (gray circle) AAUAAA, UGUA, or U/G-rich motifs individually (**b**, **c**)

or in combination (**d**, **e**). **f**, **g** Box plots (center line, median; box limits, upper and lower quartiles; whiskers, 1.5× interquartile range) and underlying data points for the strength of terminators supplemented with the indicated motifs relative to the corresponding terminator without any motif (set to 0). In (**b**–**g**), red circles indicate presence of a motif and gray circles indicate its absence. Violin plots, box plots, and significance levels in (**b**–**g**) are as defined in Fig. 1.

is found in nearly 90% of all terminators and consists of a G nucleotide surrounded by 2 to 3 U nucleotides. In contrast, the motif identified in the maize protoplast data is much less prevalent (16% of all terminators) and consists of alternating U and G nucleotides (Fig. 4a). Despite these differences, both motifs have an influence on terminator strength, leading to a 40% and 20% increase in strength for terminators with the motif as compared to those without, in tobacco and maize respectively (Fig. 4b, c).

While each individual motif contributes positively to terminator strength, we observed additive effects when multiple motifs are present within the same terminator. Independent of the assay system, the strongest terminators contain all three motifs and were on average 60% more active than terminators without any motifs (Fig. 4d, e). To validate the effect of the identified AAUAAA and UGUA motifs on terminator strength, we selected 20 terminators each with a single strong AAUAAA or UGUA motif and generated single or double nucleotide mutations to break the motif. The terminator strength of the original and mutated sequences was then measured in tobacco leaves and maize protoplasts. Most motif mutations lowered terminator strength (Supplementary Fig. 8c, d). Mutations in the AAUAAA motif caused, on average, a 24% decrease in terminator strength in tobacco leaves and a 16% decrease in maize protoplasts. Mutations in the UGUA motif decreased terminator strength by approximately 13% and 7% in tobacco leaves and maize protoplasts, respectively.

Next, we wondered if we could use the enriched motifs to improve the strength of weak terminators. We selected randomized sequences from our initial data that showed low terminator strength and did not contain an AAUAAA, a UGUA, or a U/G-rich motif. We then inserted a CA dinucleotide at the predicted cleavage site (position 150) and added all possible single, double, and triple combinations of the AAUAAA, UGUA, and U/G-rich motifs. All sequences were then subjected to Plant STARR-seq in tobacco leaves and maize protoplasts. In most cases, adding motifs increased terminator strength and additive effects were observed when multiple motifs were inserted into a terminator (Fig. 4f, g). On average, single motifs led to a 5% to 20% increase in terminator strength. Terminators with all three motifs were approximately 60% stronger than motif-less terminators in tobacco leaves and maize protoplasts. Although the GUGUG motif was originally identified in data from the maize protoplast system, it also increased terminator activity in tobacco leaves. Taken together, these findings demonstrate that the motifs enriched in strong terminators are indeed contributing to terminator strength.

## Computational models accurately predict terminator strength

Computational models can successfully predict the activity of regulatory elements by learning key sequence features[4,56,57]. To develop computational models that can predict terminator strength, we initially focused on using k-mer counts as a proxy for terminator activity. To test the validity of this approach, we counted the occurrence of all possible 6-mers (4,096 sequences) in the terminator sequences, and calculated the correlation between terminator strength and how often a given 6-mer was represented in a terminator sequence. As expected, we found that the 6-mers most correlated with terminator strength in tobacco leaves and maize protoplasts are variations of the canonical cleavage and polyadenylation signals AAUAAA and UGUA (Fig. 5a, b). However, we observed only a moderate conservation of the 6-mer rank orders between the two assay systems (Spearman's ρ = 0.29; Fig. 5c). 6-mers that were highly correlated with terminator strength in tobacco leaves but not in maize protoplasts were U-rich. Conversely, 6-mers with a high correlation to terminator strength in maize protoplasts but not in tobacco leaves were rich in G and C nucleotides (Fig. 5c, d). These findings are consistent with our prior observations on sequence motifs associated with high terminator activity, both within and across the two assay systems. Therefore, we used 6-mer counts as input features to train a lasso regression model to predict terminator strength.

The model was trained on 70% of our terminator data and tested on 30% of the data. The lasso regression model had moderate predictive power, explaining 28% of the variance in terminator strength in tobacco leaves and 41% in maize protoplasts (Fig. 5e, f).

To build a model with increased predictive power, we turned to a convolutional neural network with a DenseNet architecture[58], because this approach had worked well with Plant STARR-seq data previously[59]. Our DenseNet model uses the sequence of a terminator as an input and predicts the strength of this sequence in tobacco leaves and maize protoplasts. After the model was trained on 90% of the terminator data, it could accurately predict terminator strength for the remaining 10% of the data. The features learned by the model explained 76% and 67% of the variance in terminator strength in tobacco leaves and maize protoplasts, respectively (Fig. 6a, b). To understand what features the DenseNet model had learned, we used DeepLIFT and TF-MoDISco to extract consolidated sequence motifs that positively or negatively impact terminator strength[60,61] prediction (Fig. 6c–f). According to this analysis, AAUAAA and UGUA motifs are associated with increased terminator strength in both assay systems. U-rich sequences decrease terminator strength in maize protoplasts, especially if they surround a UGUA motif. In contrast, U-rich sequences increase terminator strength in tobacco leaves, although this effect can be reversed by GG dinucleotides (Fig. 6e, f). These findings are consistent with the results from our motif enrichment and k-mer analyses and indicate that the DenseNet model has learned biologically relevant terminator features.

## In silico evolution of terminator sequences

While our previous attempts to improve terminator strength by adding polyadenylation and cleavage motifs were successful, the gain in terminator strength was modest (Fig. 4f, g). In silico evolution using convolutional neural network models can drastically improve the activity of regulatory elements[4,56]. Therefore, we used our DenseNet model for in silico evolution of 222 terminators, 111 each from *Arabidopsis* and maize. As starting sequences, we randomly selected terminators from the held-out test set used to validate the DenseNet model. For each terminator, we generated every possible single-nucleotide substitution variant and scored these variants with the DenseNet model. We kept the variant with the highest predicted terminator strength and subjected it to another round of evolution for a maximum of ten rounds (Fig. 7a). We performed this process three times using the prediction for either tobacco leaves or maize protoplasts, or the sum of both predictions to identify the strongest variant. To test the evolved elements experimentally, we synthesized the starting sequences and those obtained after three and ten rounds of evolution, and experimentally assayed their terminator strength in tobacco leaves and maize protoplasts.

After three rounds of evolution (*i.e.*, after changing only three nucleotides), terminator strength increased, on average, 5-fold in tobacco leaves and 1.6-fold in maize protoplasts (Fig. 7b, c). For sequences obtained after ten rounds of evolution, a further increase in terminator strength was observed. Even though the original sequences and those obtained after in silico evolution differed by only ten nucleotides, the difference in their terminator strength was 12-fold in tobacco leaves and 4-fold in maize protoplasts (Fig. 7b, c). As expected, the increase in terminator strength was most pronounced when the prediction used to score variants during in silico evolution matched the experimental assay system (*i.e.* using the tobacco leaf prediction for evolution, and testing the resulting terminators in tobacco leaves). Although all starting sequences were weaker than the 35S terminator, our in silico evolution approach generated several terminators of greater strength than this highly active viral terminator (Fig. 7b, c). We tested selected evolved terminators with the dual-luciferase assay and found strong correspondence with the Plant STARR-seq results (Fig. 7d, e). Furthermore, we observed that the model favored the

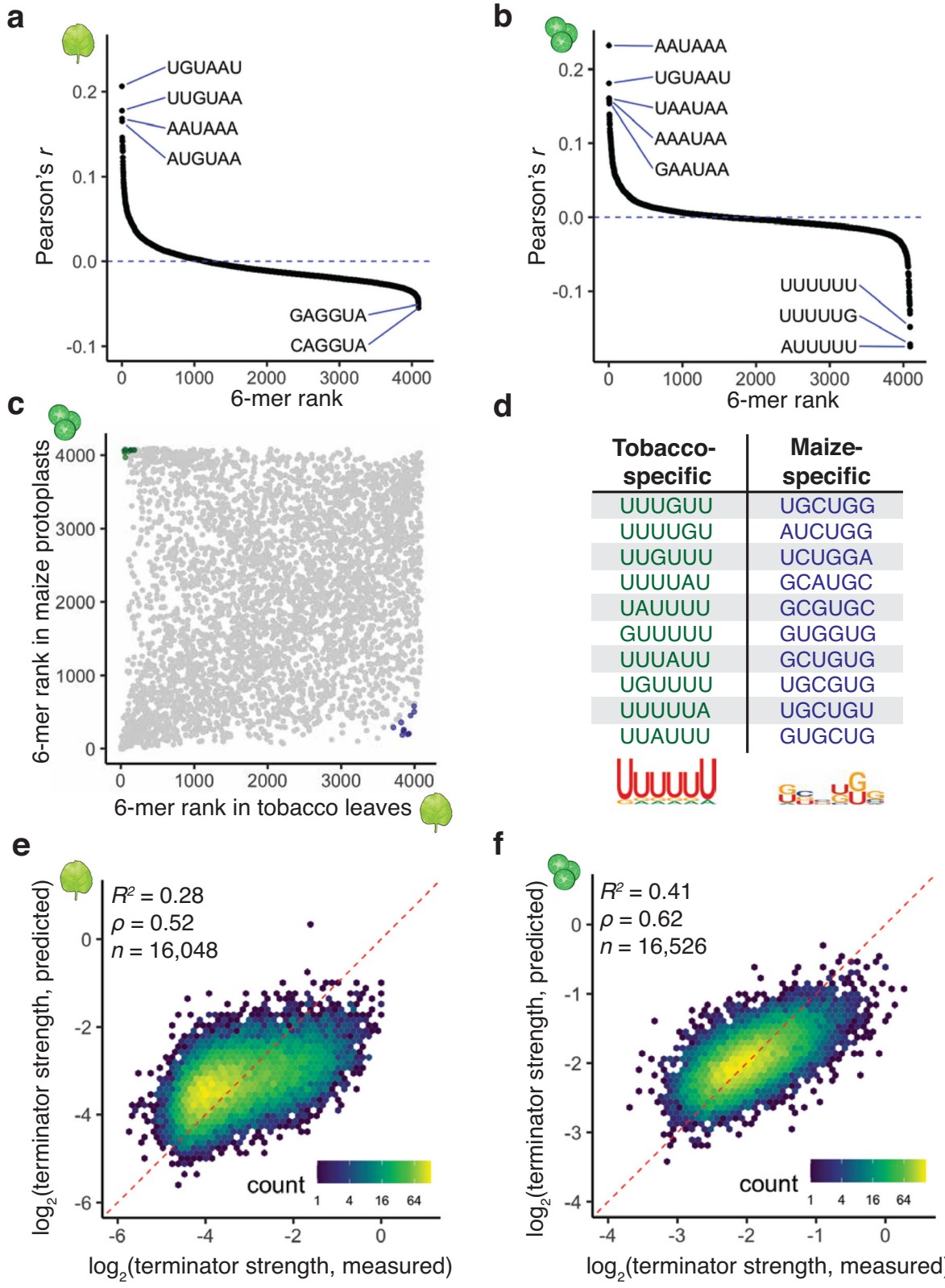

**Fig. 5 | K-mers can be used for species-specific terminator strength predictions.**
**a**, **b** Correlation (Pearson's *r*) between terminator strength in tobacco leaves (**a**) or
maize protoplasts (**b**) and how often a given 6-mer is present in the terminators.
The horizontal axis displays the 6-mer "rank" based on Pearson's correlation. The
sequences of the highest and lowest ranked 6-mers are indicated. **c** Comparison of
the 6-mer ranks in tobacco leaves and maize protoplasts. 6-mers highlighted in
green and blue had the biggest differences in rank order between the two assay

systems. **d** Top 10 tobacco- or maize-specific 6-mers (highlighted in **c**) and
sequence logo plots generated from them. **e**, **f** Hexbin plot (as defined in Fig. 1) of
the correlation between terminator strength measured by STARR-seq in tobacco
leaves (**e**) or maize protoplasts (**f**) and terminator strength predictions from a lasso
regression model based on 6-mer counts. Only the terminators not used for model
training are shown (30% of all terminators).

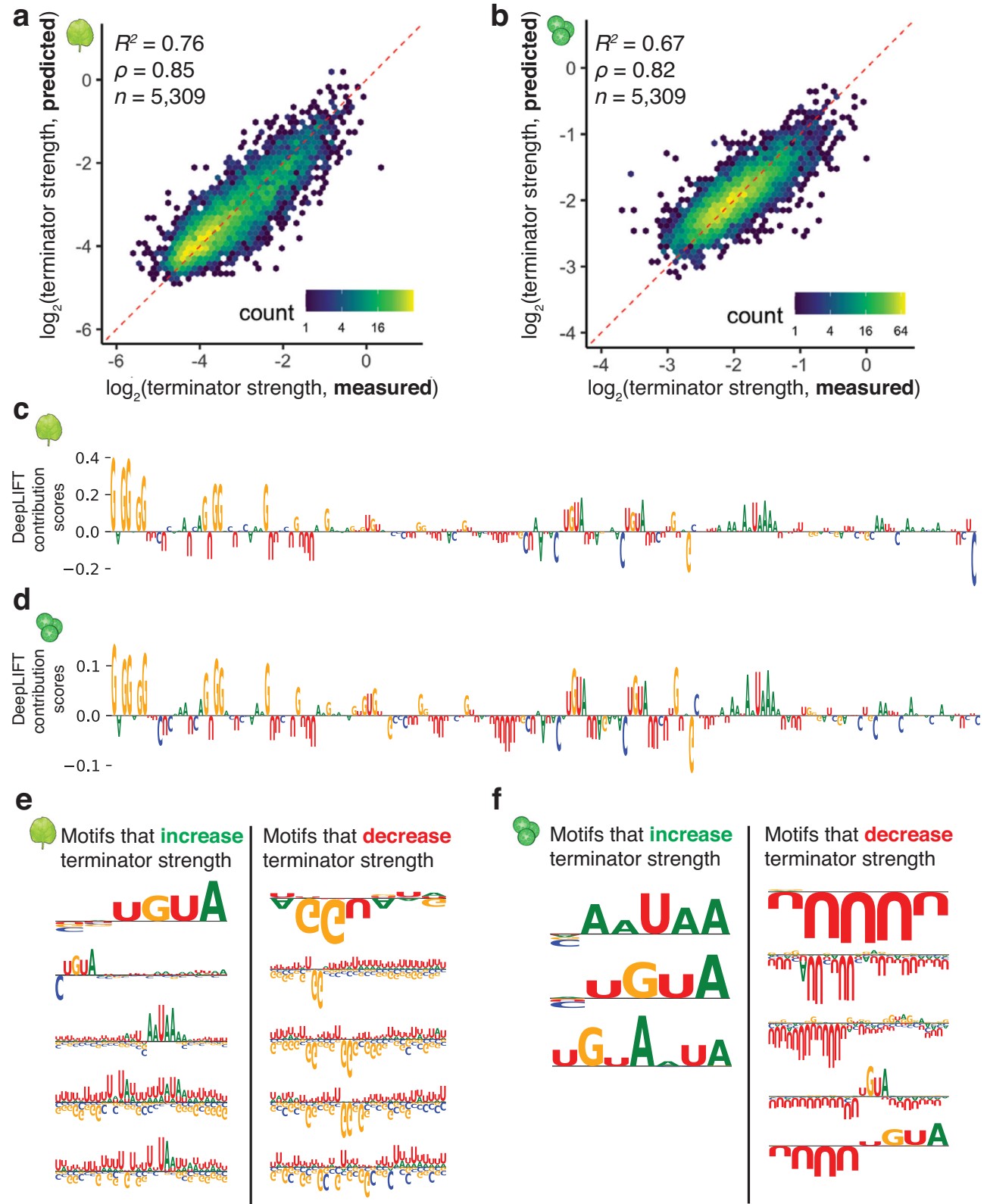

**Fig. 6 | A convolutional neural network accurately predicts terminator strength. a**, **b** Hexbin plot (as defined in Fig. 1) of the correlation between terminator strength measured by STARR-seq in tobacco leaves (**a**) or maize protoplasts (**b**) and terminator strength predictions from a DenseNet convolutional neural network trained on terminator sequences. Only the terminators not used for model training are shown (10% of all terminators). **c**, **d** DeepLIFT importance scores based on our DenseNet model predictions of terminator strength in tobacco leaves (**c**) or maize protoplasts (**d**). The terminator of AT1G31180 is shown as an example. **e**, **f** Motifs identified by TF-MoDISco that positively or negatively contribute to the DenseNet model predictions of terminator strength in tobacco leaves (**e**) or maize protoplasts (**f**).

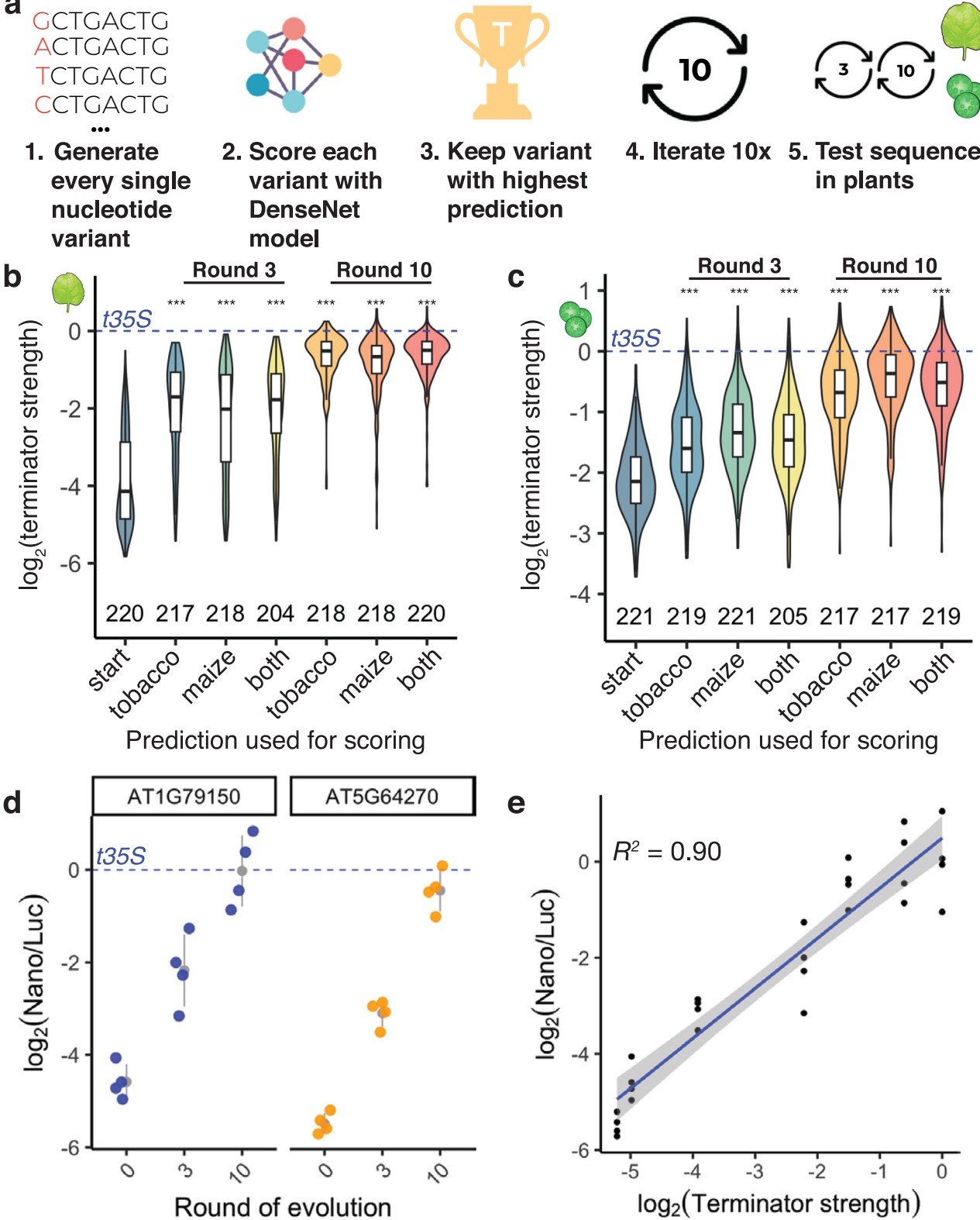

**Fig. 7 | In silico evolution of plant terminators. a** Scheme of the in silico evolution. **b**, **c** Violin plots, box plots, and significance levels (as defined in Fig. 1) of terminator strength measured in tobacco leaves (**b**) or maize protoplasts (**c**) for unmodified terminators (start) and terminators after three or ten rounds of in silico evolution. Groups are statistically compared to the "start" sequence. The DenseNet model prediction for tobacco leaves (tobacco) or maize protoplasts (maize), or the sum of both predictions (both) was used as a score during in silico evolution. The dashed blue line indicates the strength of the 35S terminator (t35S). **d** Jitter plot of the nanoluciferase activity of the in silico evolution of AT1G79150 and AT5G64270 terminators. The dashed blue line indicates the average nanoluciferase activity of the 35S terminator (t35S), set to 0. The gray dot denotes the mean and the gray line denotes the variance. **e** Dot plot and Pearson's $R^2$ between terminator strength and nanoluciferase activity of evolved terminators in (**d**). Linear regression line is shown as a blue line, and the gray band around the regression line is the 95% confidence interval. Raw values and calculated scores from assay are provided in Supplementary Data 5.

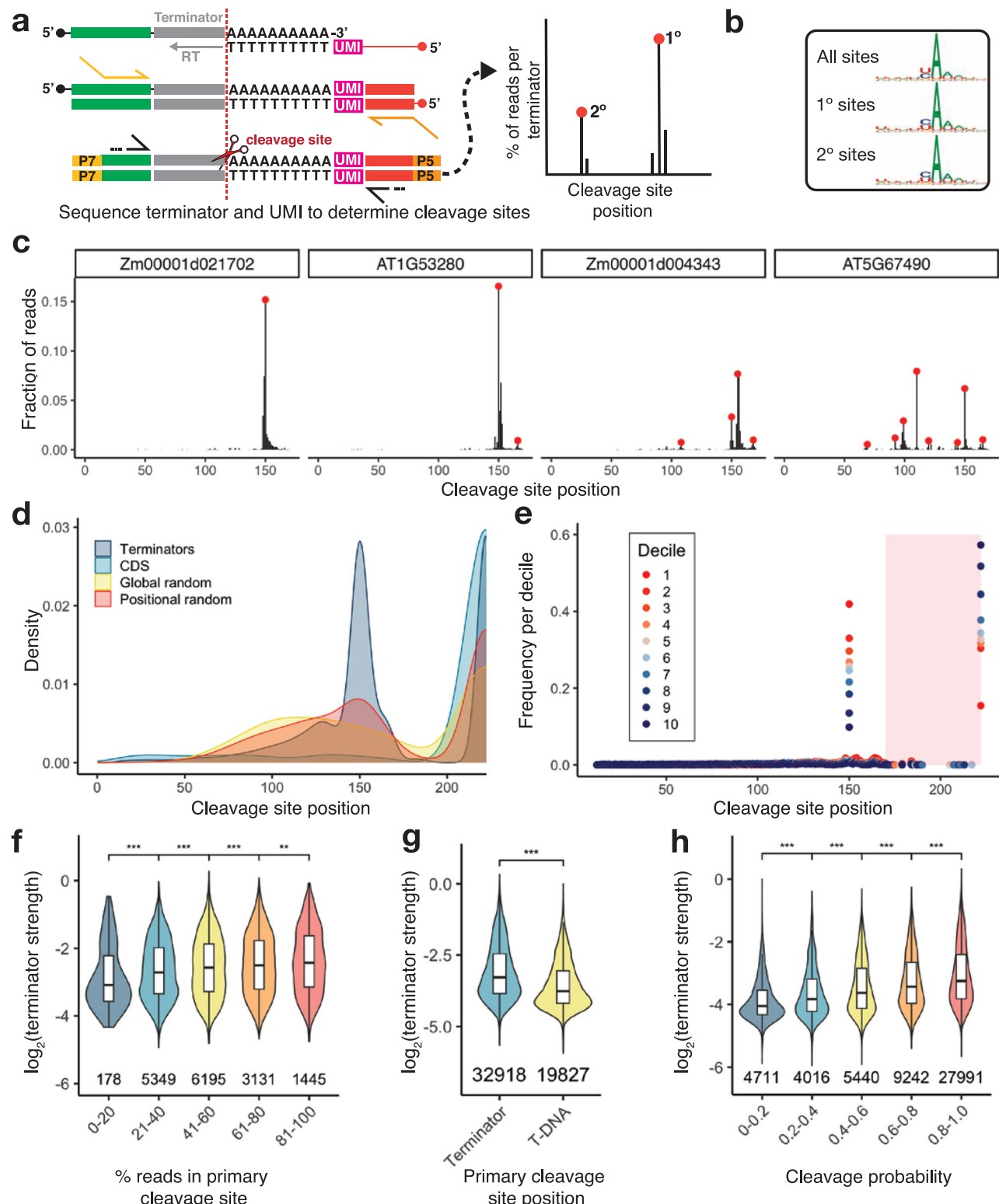

insertion of multiple UGUA motifs into the evolved, stronger terminators (Supplementary Fig. 8g).

## Cleavage and polyadenylation efficiency affects terminator strength

Over 70% of the genes in *Arabidopsis* and maize contain more than one possible polyadenylation site and can therefore give rise to transcript isoforms that differ in 3' UTR length[42,44]. Alternative polyadenylation can have a strong effect on transcript levels through the

inclusion or exclusion of regulatory features such as binding sites for microRNAs or RNA-binding proteins[41]. We wondered whether the terminator strength of putative polyadenylation sites affects the choice of which site is used. To address this question, we measured the terminator strength of primary and secondary cleavage and polyadenylation sites from over 5000 genes. However, there was no difference between the terminator strength of primary and secondary polyadenylation sites for *Arabidopsis* or maize terminators (Supplementary Fig. 9).

**Fig. 8 | Polyadenylation and cleavage affect terminator strength.**
**a** Polyadenylation and cleavage sites for terminators in tobacco leaves were determined by 3′-end sequencing. Using an oligo-dT primer with a unique molecular identifier (UMI), cDNA was generated from polyadenylated terminators. The cDNA was subjected to paired-end sequencing of the terminator and UMI. Cleavage sites were defined as local maxima in a histogram of terminator read length after trimming of the poly-A tail. **b** Sequence logo plots generated from the 10 bp window around all cleavage sites (All sites), all primary sites (1° sites), or all secondary sites (2° sites). **c** Histograms of cleavage site position for four representative terminators. Red dots denote primary and alternative cleavage sites. **d** Kernel density distribution of the primary cleavage site position for plant terminators (Terminators), sequences from coding regions (CDS), and randomized sequences with an overall (Global random) or per-position (Positional random) nucleotide frequency similar to an average *Arabidopsis* or maize terminator. **e** Histogram of the primary cleavage site position for terminators grouped into deciles based on terminator strength in tobacco leaves (decile 1 contains the strongest 10% of the terminators and decile 10 the weakest 10%). Each decile contains approximately 5300 terminators. The shaded red area corresponds to cleavage in the T-DNA backbone (*i.e.* downstream of the terminator sequence). **f** Violin and box plots of terminator strength. Terminators were grouped by the percentage of reads that coincide with the primary cleavage site. **g** Violin plots of the strength of terminators with a primary cleavage site within the terminator or in the T-DNA. **h** Violin plots of terminator strength for terminators grouped by cleavage probability (percent of unique reads showing cleavage within the terminator sequence). Violin plots, box plots, and significance levels in (**f**–**h**) are as defined in Fig. 1. Major cleavage and polyadenylation sites per terminator are reported in Supplementary Data 6.

**Table 1 | Terminator polyadenylation and cleavage statistics**

| Terminator group | Cleavage site at position 150 ± 5 bp | Primary cleavage site at position 150 ± 5 bp | Multiple cleavage sites | Strong secondary cleavage site (≥30% of reads) | Cleavage in T-DNA backbone only |
|---|---|---|---|---|---|
| All terminators n = 54,825 | 33,700 (61%) | 17,986 (32%) | 37,301 (68%) | 7393 (13%) | 9525 (17%) |
| *Arabidopsis* terminators n = 22,587 | 15,331 (68%) | 8826 (39%) | 16,827 (74%) | 4421 (20%) | 1082 (5%) |
| Maize terminators n = 27,733 | 16,931 (61%) | 8,540 (31%) | 18,008 (65%) | 2474 (9%) | 6879 (25%) |
| CDS n = 936 | 36 (4%) | 6 (1%) | 234 (25%) | 17 (2%) | 663 (71%) |
| Global random sequences n = 381 | 68 (18%) | 15 (4%) | 227 (60%) | 53 (14%) | 107 (28.1%) |
| Positional random terminators n = 1865 | 773 (41)% | 285 (15%) | 1,250 (67%) | 266 (14%) | 388 (21%) |

While terminator strength could not predict which potential polyadenylation site is more likely to be used in a genomic context, we wondered if the reverse were true: Is the strength of a terminator associated with the efficiency of its cleavage and polyadenylation? To determine the cleavage and polyadenylation sites in our terminators, we subjected the output RNA from a Plant STARR-seq experiment in tobacco leaves to 3′-end sequencing (Fig. 8a). We defined the cleavage and polyadenylation sites for each terminator as local maxima in the distribution of the cleavage positions along the terminator sequence for all unique reads. As expected, the vast majority of the cleavage and polyadenylation sites identified by this approach coincided with a CA or UA dinucleotide (Fig. 8b) matching the known cleavage element[10,13]. Consistent with prior reports[15,62–64] we observed near complete cleavage of the 35S terminator with one major cleavage site at the same position as previously shown (Supplementary Fig. 10b). We confirmed the cleavage pattern of several terminators with RT-PCR and gel electrophoresis (Supplementary Fig. 10c).

The terminators in our library were designed to contain a cleavage site at position 150 (Fig. 1a). About 61% of the terminators showed a cleavage site within 5 nucleotides upstream or downstream of this position, and for approximately half of these (32% of all terminators), this was the primary cleavage site (Table 1). Of the 54,825 terminators for which we could call cleavage sites, 68% had more than one cleavage sites, and of those, 20% (about 13% of all terminators) had a prominent secondary cleavage site indicated by at least 30% of the reads (Table 1; see Fig. 8c for examples). However, 17% of the tested sequences were not cleaved and polyadenylated but showed read-through into the downstream T-DNA sequence.

We observed differences in the cleavage pattern between bona fide terminators and control sequences (Fig. 8d, Table 1). Plant terminators were predominantly cleaved at position 150. In contrast, randomized sequences showed a broad distribution of cleavage sites throughout their sequence. However, for the positional random

sequences which resemble terminators more closely than the global random controls, the distribution was shifted towards an enrichment of cleavage at or near position 150. Finally, control sequences derived from coding regions were rarely cleaved and polyadenylated, consistent with a selection against potential cleavage sites. These general trends held true when considering all cleavage sites, in addition to primary sites (Supplementary Fig. 10a).

Next, we tested the association of terminator strength and cleavage probability (*i.e.*, the likelihood of a terminator being cleaved and polyadenylated instead of allowing read-through transcription). Cleavage probability was strongly correlated with terminator strength (Fig. 8e, h), and terminators with a primary cleavage site within their sequence were approximately 40% stronger than those without (Fig. 8g). Furthermore, terminators with a dominant cleavage site (*i.e.*, most transcripts are cleaved at this site) were stronger terminators than those with multiple weak cleavage sites (Fig. 8f). Taken together, the efficiency and accuracy with which a terminator is cleaved and polyadenylated determines its strength. Strong terminators show a dominant cleavage site and prevent read-through into downstream sequences.

**Polyadenylation motifs and GC content control cleavage probability**

Since cleavage probability and terminator strength were positively correlated, we asked if the same features that influence terminator strength also affect cleavage probability. First, we tested if biological terminator function was associated with a high cleavage probability. Indeed, plant terminators showed a much higher cleavage probability than control sequences from coding regions (Fig. 9a). Randomized control sequences with a nucleotide composition similar to an average terminator were also significantly more likely to be cleaved and polyadenylated than the coding region controls, but their average cleavage probability was still lower than that of bona fide plant terminators. The

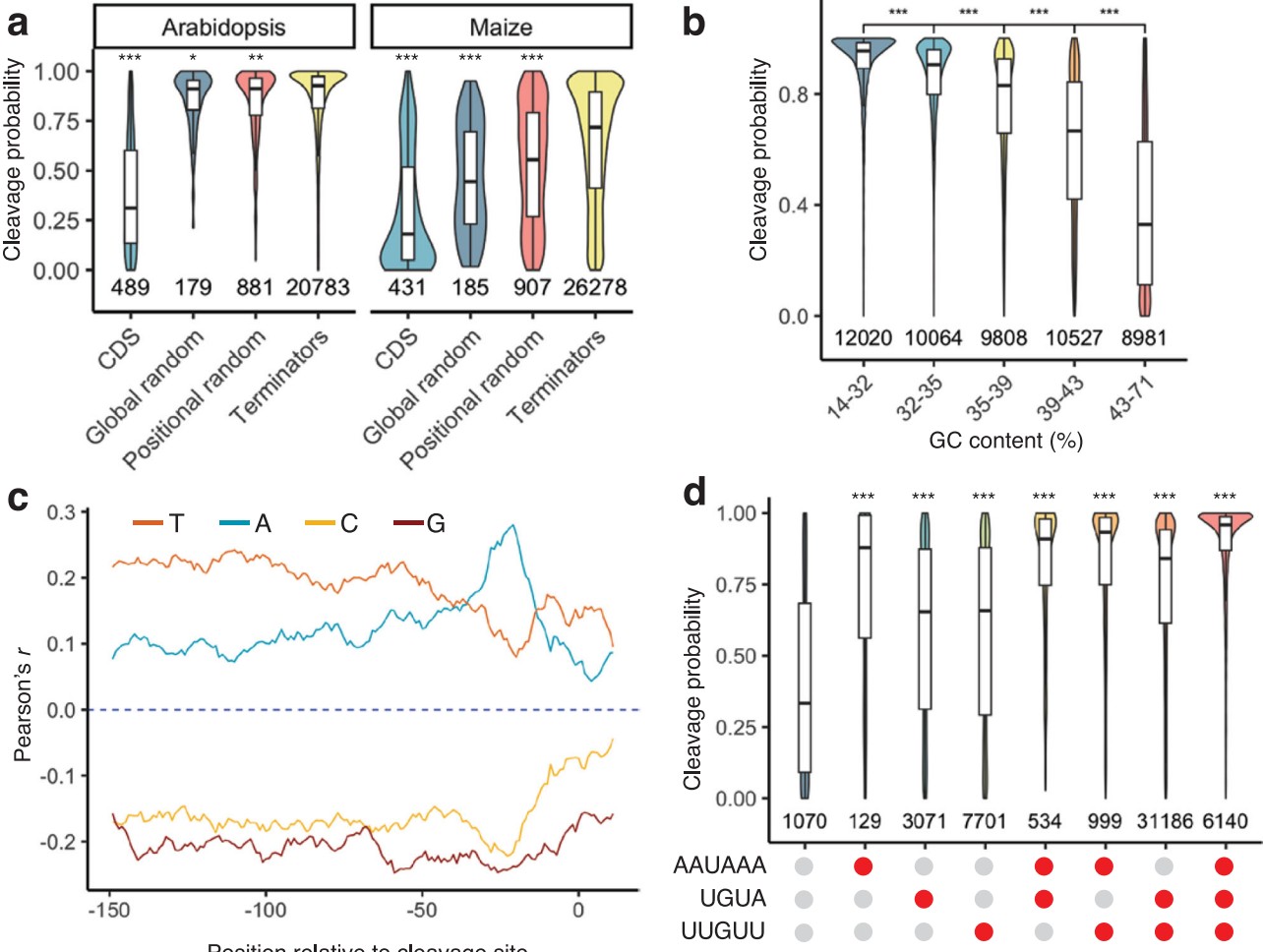

**Fig. 9 | GC content and polyadenylation motifs influence cleavage probability.** **a** Violin plots, box plots, and significance levels of cleavage probability (percent of unique reads showing cleavage within the terminator sequence) for plant terminators (Terminators) compared to sequences from coding regions (CDS) and randomized sequences with an overall (Global random) or per-position (Positional random) nucleotide frequency similar to an average *Arabidopsis* or maize terminator. Each control group is correlated to the terminator group. **b** Violin plots, box plots, and significance levels of cleavage probability. Terminators were binned by GC content to yield groups of approximately the same size. **c** Correlation (Pearson's *r*) between cleavage probability and the A, C, G, or U content of a ten-base window starting at the indicated position in the terminators. **d** Violin plots, box plots, and significance levels of cleavage probability for terminators with (red dot) or without (gray dot) the indicated motifs. Each group is correlated to the no motif control. Violin plots, box plots, and significance levels in (**a**, **b**, **d**) are as defined in Fig. 1.

cleavage probability for sequences derived from *Arabidopsis* was higher than the one for maize sequences (Fig. 9a).

As GC content is one of the key differences between terminators from *Arabidopsis* and maize, we asked if cleavage probability was affected by GC content. We observed a strong negative correlation (Pearson's $r = -0.65$) between these two factors. While a high AT content was associated with a high cleavage probability, increasing GC content led to decreased cleavage probability (Fig. 9b). This finding likely explains why maize sequences, which show on average an approximately 10% higher GC content than *Arabidopsis* sequences, were less likely to be cleaved than *Arabidopsis* sequences. Our observation that AU-rich sequences show high cleavage probability is consistent with previous reports that the insertion of AU-rich sequences into transcripts can lead to their cleavage and polyadenylation in maize[65]. Although both terminator strength and cleavage probability were affected by GC content, they followed different trends. Cleavage probability decreased monotonically with increasing GC content, while terminator strength was highest at an intermediate GC content and lowest in very AT- or GC-rich sequences (Fig. 3b, c). Similarly, we observed differing and position-specific effects of local nucleotide composition on terminator strength, whereas for cleavage probability,

positional effects were much less pronounced, and A and T nucleotides behaved mostly similar to each other as did G and C nucleotides (Fig. 9c).

Finally, we tested if the AAUAAA, UGUA and U/G-rich motifs affected cleavage probability. While all three motifs increased cleavage probability, the AAUAAA motif clearly had the largest effect (Fig. 9d). As for terminator strength, we observed additive effects of the motifs on cleavage probability, and terminators with all three motifs were most likely to be cleaved and polyadenylated.

## Discussion

Here, we adapted Plant STARR-seq to characterize and optimize plant terminator sequences. Our results capture the effects of these sequences on RNA expression, stability and cleavage probability. We find that thousands of *Arabidopsis* and maize terminators outperformed the commonly used bacterial NOS and MAS terminators; a handful even outperformed the viral 35S terminator. These results provide a large arsenal of diverse and strong plant terminators to use with transgenes, which should alleviate transgene silencing[22]. The optimal terminators to employ in constructing transgenes may be those with high cleavage probability, as these prevent the

read-through transcription implicated in silencing[23]. Thus, for use in dicots, good terminators to choose would be those from the *Arabidopsis* genes AT3G46230 (HEAT SHOCK PROTEIN 17.4), AT2G05530, and AT4G39730 (PLAT DOMAIN PROTEIN 1); for use in monocots, good ones would be those from the maize genes Zm00001d016542 (anthranilate 1,2-dioxygenase), Zm00001d047961, and Zm00001d017119 (glucose-6-phosphate dehydrogenase 5).

In addition to these naturally occurring terminators, in silico evolved, synthetic terminator sequences − several of which showed greater strength and higher reporter protein abundance than the viral 35S terminator − resulted from this work. Use of the DenseNet prediction that best matched the assay system (the tobacco leaf prediction for dicots, and the maize protoplast prediction for monocots) led to the best results; however, a combination of both predictions can generate terminators that are active across a broad range of plant species, tissues, and applications. The iterative in silico evolution and testing of evolved elements in Plant STARR-seq appears to be a promising generalizable path toward improving the features of plant regulatory elements, as for core promoters[4,66].

Beyond the practical benefits of this study, we identified novel features of terminator biology in plants. We found that GC content profoundly influences terminator strength and cleavage probability in different ways. The strongest terminators had a GC content of approximately 30–40% (depending on the assay system), with terminator strength lowest at either higher or lower GC percentage in both assay systems. In contrast, cleavage probability monotonically decreased with increasing GC content in tobacco leaves (and was not measured in maize protoplasts due to technical challenges). This negative correlation is similar to the effects of GC content on core promoter strength, which shows a strong negative correlation with GC content in tobacco leaves but not in maize protoplasts[4]. Taken together, GC content appears to affect different regulatory elements in different ways, and these effects vary across species, reflecting genomic GC content.

In addition to known polyadenylation motifs[10,12,41], we discovered motifs that increased terminator strength and cleavage probability. These motifs are rich in U and G nucleotides and reside broadly across terminator sequences, with a moderate enrichment immediately upstream and downstream of the cleavage site. The precise composition of these U/G-rich motifs differed between the tobacco leaf and the maize protoplast systems; however, the maize-specific motif also contributed to terminator strength in tobacco leaves. Human terminators contain a GU-rich element downstream of the cleavage site that is required for efficient polyadenylation and that is bound by the cleavage-stimulating factor (CSTF) complex[14,67,68]. Plant homologs for members of the CSTF complex can form a similar complex and bind to RNA[69–71]. We speculate that the U/G-rich motifs identified here serve as binding sites for the plant CSTF complex.

The assay design employed here provided insight into cleavage probability and alternative polyadenylation. Alternative polyadenylation is a crucial mechanism to create diverse transcripts in response to environmental change and in particular tissues[41,44,72]. Alternative polyadenylation produces transcripts with altered 3′ UTR length, resulting in different mRNAs and translation levels. Long 3′ UTRs decrease transcript stability by triggering nonsense-mediated mRNA decay[73], while short 3′ UTRs make translation more efficient by promoting polysome formation[74]. When we compared the 170 nucleotide-terminator sequences of *Arabidopsis* and maize genes derived from primary vs. secondary polyadenylation sites, we observed no difference in terminator strength among sites derived from *Arabidopsis* or maize. These comparisons suggest that the sequence surrounding a given polyadenylation site does not fully explain alternative polyadenylation. *Trans*-acting factors such as the components of the cleavage and polyadenylation complexes, RNA-binding proteins and non-coding RNAs are involved in the choice of polyadenylation sites,

presumably by recognizing sequence context and secondary structure of the entire terminator sequence instead of the shorter terminator sequences tested here[41,75,76]. A comparison of all tested terminator sequences indicates that strong terminators tended to contain a dominant cleavage site rather than multiple weaker ones.

## Methods

### Library design and construction

For this study, we defined terminators as the sequence from 150 nucleotides upstream to 20 nucleotides downstream of a cleavage and polyadenylation site. For *Arabidopsis*, we used experimentally determined cleavage and polyadenylation sites described by ref.[43]. We selected the primary cleavage and polyadenylation sites for each gene ($n = 18,450$) in wild-type plants, as well as prominent secondary cleavage sites that had at least 30% of total reads per gene ($n = 2325$). For the 3754 genes without an experimentally defined cleavage site, we utilized the end of their 3′ UTR annotation in the *Arabidopsis* TAIR10 annotation as the cleavage site. This yielded a total of 24,529 *Arabidopsis* terminator sequences (Supplementary Table 1). No additional filtering was applied, leading to the inclusion of some coding sequences. One example is the tested HSP18.2 sequence, which differs from the strong terminator sequence described in prior studies[38] For maize, we used cleavage and polyadenylation sites defined by ref.[42]. We selected primary cleavage sites for 25,685 maize genes, and included 4407 secondary cleavage sites (at least 30% of total reads per gene) for a total of 30,092 maize terminator sequences (Supplementary Table 1).

In addition to bona fide terminators, several control sequences were added to the library (Supplementary Table 1). Since protein coding sequences should not contain cleavage and polyadenylation motifs, 170-nucleotide long sequences (589 each from *Arabidopsis* and maize) from coding regions with a similar GC content as the terminator library were included as negative controls. Furthermore, we generated random sequences with global (200 sequences per species) or position-specific (1,000 sequences per species) nucleotide frequencies derived from the average *Arabidopsis* or maize terminator. Randomized sequences with a GC content of 30%, 40%, 50%, 60%, or 70% (200 sequences of 170 nucleotides each) were added to test effects of GC content on terminator strength. Finally, four commonly used non-plant terminators (t35S, tAg7, tNOS, and tMAS) were added as a frame of reference in their full length (35S = 204 nt, Ag7 = 208 nt, MAS = 253 nt, NOS = 253 nt). The final library of sequences was ordered as an oligo pool from Twist Biosciences (Supplementary Data 1).

To validate and extend our findings from the large-scale terminator library, we created a second, smaller library. This library included 187 terminators from the original library to test the correlation between results from the two libraries. Additionally, we added sequences to validate the importance of the discovered motifs. We picked 20 sequences from the randomized control sequences in our original library that had neither a CA dinucleotide at the predicted cleavage site, nor an AAUAAA, a UGUA, or a GUGUG motif. For each validation sequence, we added a CA dinucleotide at position 150 and all possible combinations of these three motifs at the positions where they are found most often in our library. To test the importance of the AAUAAA motif, we picked twenty terminators that had only one AAUAAA sequence and generated three variants with a mutation in this sequence (ACTCAA, ACTCAA, ACTAAA). Similarly, we took twenty sequences that had only one UGUA sequence and replaced it with mutated variants (UGCA, CGUA, UCUA). Finally, we included the sequences from the in silico evolution (see below) in this validation library (Supplementary Data 2).

The Plant STARR-seq plasmid pPSt (https://www.addgene.org/203590/) used in this study is based on the pPSup plasmid (https://www.addgene.org/149416/)[5] with adaptations to enable the characterization of terminators. The base plasmid pPSt contains three consecutive Golden Gate cloning sites. To generate the terminator libraries,

we first inserted the 35S promoter together with a 5′ UTR derived from the maize histone H3 gene Zm00001d041672, an ATG start codon, and a 18-nucleotide random barcode (VNNVNNVNNVNNVNN; V = A, C or G) into the upstream Golden Gate site using the restriction enzyme BbsI-HF (NEB). Next, we cloned the array-synthesized terminator library into the downstream Golden Gate site using the restriction enzyme Esp3I (NEB). In this step, the plasmid library was bottlenecked to approximately 1,000,000 variants (approximately 16 barcodes per terminator) or 60,000 variants (approximately 30 barcodes per terminator) for the large-scale or the validation library, respectively. Finally, we used BsaI-HFv2 (NEB) to replace the central Golden Gate site with the coding sequence of GFP lacking the start codon. Two versions of the library were created in this step: one with full-length GFP for use in Plant STARR-seq (see https://www.addgene.org/203592/ for an example of a fully assembled Plant STARR-seq plasmid with the 35S terminator.) and one with a truncated version of GFP lacking the central 188 amino acids for subassembly (see below). The sequences of all primers used for cloning and sequencing are included in Supplementary Table 2.

## Tobacco cultivation and transformation

Tobacco (*Nicotiana benthamiana*) was grown in soil (Sunshine Mix no. 4) at 25 °C with a long day photoperiod (16 h light and 8 h dark; cool-white fluorescent lights Philips TL-D 58 W/840; intensity 300 μmol m$^{-2}$ s$^{-1}$). Plants were transformed approximately 3 weeks after germination. The *Agrobacterium* transformation and induction method was previously described in ref. 4. The terminator libraries were introduced into *Agrobacterium tumefaciens* strain GV3101 (harboring the virulence plasmid pMP90 and the helper plasmid pMisoG) by electroporation. *Agrobacterium* cultures were infiltrated into the first two mature leaves of tobacco plants (12 plants/24 leaves per replicate for the large-scale library; 3 plants/6 leaves for the smaller validation library). The plants were further grown for 48 hours under normal light conditions before mRNA extraction. The input sample for Plant STARR-seq was obtained from 20 mL of the *Agrobacterium* solution that was used to infiltrate tobacco leaves. To this end, the QIAprep Spin Miniprep Kit was used according to the manufacturer's instructions.

## Maize mesophyll protoplast generation and PEG transformation

We used the PEG transformation method of maize mesophyll protoplasts as described in ref. 77. Maize (Zea mays L. cultivar B73) seeds were soaked in water overnight at 25 °C. The seeds were germinated in soil for 3 days under long day conditions (16 h light, 8 h dark) at 25 °C, then moved to complete darkness at 25 °C for 10–11 days. From each seedling, 10 cm sections from the second and third leaf were cut into thin 0.5 mm strips perpendicular to veins and immediately submerged in 10 ml of protoplasting enzyme solution (0.6 M mannitol, 10 mM MES ph 5.7, 15 mg/ml cellulase R10, 3 mg/ml macerozyme, 1 mM CaCl$_2$, 0.1% [w/v] BSA, and 5 mM beta-mercaptoethanol). The mixture was covered in foil to keep out light, vacuum infiltrated for 3 min at room temperature (RT), and incubated on a shaker at 40 rpm for 2.5 h at RT. Protoplasts were released by incubating an extra 10 min at 80 rpm. To quench the reaction, 10 mL ice-cold MMG (0.6 M Mannitol, 4 mM MES ph 5.7, 15 mM MgCl2) was added to the enzyme solution and the whole solution was filtered through a 40 μM cell strainer. To pellet protoplasts, the filtrate was split into equal volumes of no more than 10 mL in chilled round-bottom glass centrifuge vials and centrifuged at 100 x g for 4 min at RT. Pellets were resuspended in 1 mL cold MMG each and combined into a single round-bottom vial. To wash, MMG was added to make a total volume of 5 mL and the solution was centrifuged at 100 x g for 3 min at RT. This wash step was repeated two more times. The final pellet was resuspended in 1–2 mL of MMG. A sample of the resuspended protoplasts was diluted 1:20 in MMG and used to count the number of viable cells using Fluorescein Diacetate as a dye.

For each replicate, 20 million protoplasts were mixed with 200 μg of the terminator plasmid library in a fresh tube, topped with MMG to a volume of 2288 μL, and incubated on ice for 30 min. For PEG transformation, 2112 μL of PEG solution (0.6 M Mannitol, 0.1 M CaCl2, 25% [w/v] poly-ethylene glycol MW 4000) was added to reach a final concentration of 12% (w/v) PEG. The mixture was incubated for 10 min in the dark at RT. After incubation, the transformation solution was diluted with 22 mL (5 × 4.4 mL) incubation solution (0.6 M Mannitol, 4 mM MES pH 5.7, 4 mM KCl), and centrifuged at 100 x g for 4 min at RT. For the PEG transformation with the validation library, we used 4 million protoplasts, 40 μg of the plasmid library, and one fifth of the buffer volumes used for the large-scale library.

After transformation, the protoplast pellet was washed with 5 mL of incubation solution, centrifuged at 100 x g for 3 min at RT, and resuspended in incubation solution to a concentration of 500 cells/μL. Protoplasts were incubated overnight in the dark at RT to allow for transcription of the plasmid library and then pelleted (4 min, 100 x g, RT). The pellet was washed with 5 mL incubation solution and centrifuged (3 min, 100 x g, RT). The pellet was finally resuspended in 5 mL incubation solution. An aliquot of the solution was used to check transformation efficiency under a microscope (12.4% [replicate 1] and 49.5% [replicate 2] transformation efficiency for the large-scale library; 62% [replicate 1] and 45% [replicate 2] for the validation library). Cells were pelleted (4 min, 100 x g, RT) and resuspended in 2 mL Trizol for subsequent mRNA extraction. An aliquot of the plasmid library used for PEG transformation was used as the input sample for Plant STARR-seq.

## Plant STARR-seq assay

For all Plant STARR-seq experiments, two independent biological replicates were performed. Different plants and fresh *Agrobacterium* cultures were used for each biological replicate.

Tobacco leaves were harvested 2 days after infiltration and partitioned into batches of 4 (large-scale library) or 3 (validation library) leaves. The leaf batches were frozen in liquid nitrogen, finely ground with mortar and pestle, and immediately resuspended in 12 mL Trizol. The suspensions were cleared by centrifugation (5 min, 4000 x g, 4 °C) and each supernatant was mixed with 2.5 mL chloroform. After centrifugation (15 min, 4000 x g, 4 °C), 7 mL of the upper, aqueous phase was transferred to a new tube, and mixed by inversion with 3.5 mL high salt buffer (0.8 M sodium citrate, 1.2 M NaCl) and 3.5 mL isopropanol. The solution was incubated for 15 min at RT to precipitate the RNA and centrifuged (30 min, 4000 x g, 4 °C). The pellet was washed in 10 mL ice-cold 70% ethanol, centrifuged (5 min, 4000 x g, 4 °C), and air-dried. The pellet was resuspended in 180 μL of warm (65 °C) nuclease-free water and transferred to a new tube. The solution was supplemented with 10 μL 20X DNase I buffer (1 mM CaCl$_2$, 100 mM Tris pH 7.4), 10 μL 200 mM MnCl2, 2 μL DNase I (ThermoFisher Scientific), and 1 μL RNaseOUT (ThermoFisher Scientific), and incubated for 30 min at 37 °C. To precipitate the RNA, 20 μL 8 M ice-cold LiCl and 500 μL ice-cold 100% ethanol was added. After incubation for 15 min at −80 °C, the RNA was pelleted by centrifugation (20 min, 20,000 x g, 4 °C). The pellet was washed with 500 μL ice-cold 70% ethanol, centrifuged (5 min, 20,000 x g, 4 °C), air-dried for 10 min, and resuspended in 200 μL nuclease-free water supplemented with 0.5 μL RNaseOUT. For cDNA synthesis, eight reactions with 11 μL mRNA solution, 1 μL 2 μM GFP-specific reverse transcription primer, and 1 μL 10 mM dNTPs were incubated at 65 °C for 5 min then immediately placed on ice. The reactions were supplemented with 4 μL 5X SuperScript IV buffer, 1 μL 100 mM DTT, 1 μL RNaseOUT, and 1 μL SuperScript IV reverse transcriptase (ThermoFisher Scientific). To ensure that the samples were largely free of DNA contamination, four reactions were used as controls, where the reverse transcriptase and RNaseOUT were replaced with water. Reactions were incubated for 10 min at 55 °C, followed by 10 min at 80 °C. Sets of 4 reactions each were pooled. The cNDA was

purified with the Zymo Clean&Concentrate-5 kit, and eluted in 20 μL 10 mM Tris. The barcode was amplified with 10-20 cycles of polymerase chain reaction (PCR) and read out by next generation sequencing.

For Plant STARR-seq in maize protoplasts, the protoplast-containing Trizol solution from PEG transformation was transferred to 2 mL Phasemaker tubes (1 mL per tube; ThermoFisher Scientific), mixed thoroughly with 300 μL chloroform, and centrifuged (5 min, 15,000 x g, 4 °C). RNA was extracted using the RNeasy Plant Mini Kit (QIAGEN). The supernatant was transferred to a QIAshrededer column and centrifuged (2 min, 20,000 x g, RT). The flowthrough was transferred to a new 1.5 mL tube and mixed with 300 μL 100% ethanol. Up to 500 μL of the solution was loaded on an RNeasy mini spin column. After centrifugation (10 s, 16,100 x g, RT) the flowthrough was discarded. This was repeated until all the solution had been added to the column. The column was washed once with 700 μL RW1 buffer and twice with 500 μL RPE buffer. After each wash step, the column was centrifuged (30 s, 16,100 x g, RT) and the flowthrough was discarded. The column was dried with an extra centrifugation step (30 s, 16,100 x g, RT) and transferred to a 1.5 mL collection tube. For elution, 50 μL of RNase-free water was added, and the column was incubated for 1 min, and centrifuged (1 min, 16,100 x g, RT). This elution step was repeated with an additional 40 μL of RNase-free water. The eluate was treated with DNase I (5 μL of 20x DNaseI buffer, 5 μL 200 mM MnCL$_2$, 1 μL RNaseOUT, and 2 μL DNase I) for 1 h at 37 °C. The solution was supplemented with 20 μL 500 mM EDTA, 1 μL 20 mg/mL glycogen, 12 μL ice-cold 8 M LiCl, and 300 μL ice-cold 100% ethanol. The solution was incubated 15 min at −80 °C, centrifuged (20 min, 20,000 x g, 4 °C). The pellet was washed with 500 μL ice-cold 70% ethanol, and centrifuged (3 min, 20,000 x g, 4 °C). The pellet was air-dried for 10 min and resuspended in 100 μL RNase-free water. Reverse transcription, purification, PCR amplification and sequencing were performed as for the tobacco samples.

### Subassembly and barcode sequencing
Paired-end sequencing on an Illumina NextSeq 2000 platform was used to link terminators to their respective barcodes. To facilitate sequencing, we created a shortened version of the terminator library plasmid with a large deletion in the GFP gene. The terminator region was sequenced using paired 151-nucleotide reads, and two 18-nucleotide indexing reads were used to sequence the barcodes. The paired terminator and barcode reads were assembled using PANDAseq (version 2.11)[78] and the terminator reads were aligned to the designed terminator library using BowTie2 (version 2.4.1)[79]. Terminator-barcode pairs with less than 5 reads and terminators with a mutation or truncation were discarded. For each Plant STARR-seq experiment, barcodes were sequenced using paired-end reads on an Illumina NextSeq 2000 system. The paired barcode reads were assembled using PANDAseq.

### Computational methods
For calculating terminator strength, the reads for each barcode were counted in the input and cDNA (output) samples. Barcodes with less than five reads were discarded. For each terminator, the sum of the reads for all associated barcodes was calculated in the input and output samples. The input and output counts were normalized by dividing each by the sum of all counts in the output and input sample, respectively. Terminator strength was calculated as the normalized output counts divided by the normalized input counts. Terminator strength was normalized to the strength of the 35S terminator. We used the average terminator strength across two replicates for all analyses (Supplementary Data 1 and 2). Spearman and Pearson's correlation were calculated using base R (version 4.3.0). Significance was calculated using the two-sided Wilcoxon rank-sum test and the ggsignif library (version 0.6.4) in R (Supplementary Data 3). GO term

enrichment analysis was performed using the ggprofiler2 package (version 0.2.1) in R (Supplementary Data 4).

### Discovery of terminator motifs
To find motifs enriched in strong terminators, we ranked terminators according to their strength in tobacco leaves or maize protoplasts. The sequences of the top 25% terminators were analyzed by STREME (version 5.5.1)[54] to find ungapped RNA motifs that are enriched in this set relative to the sequences of the bottom 25% terminators. To find tobacco-specific terminator motifs, we used STREME with the same parameters but using the top 10% *Arabidopsis* of terminators ranked by Ψ as positive and the bottom 10% of maize terminators ranked by Ψ as negative sequences. For maize-specific terminator motifs, we repeated the analysis but switched the positive and negative sequences. Meme files with all discovered motifs are available on GitHub (https://github.com/lampoona/Terminators-Plant-STARR-seq).

All terminator sequences were analyzed using the universalmotif package (version 1.18.0) in R to find the position and frequency of the discovered motifs. For each sequence, the maximum motif score was identified and normalized to the minimum (set to 0) and maximum (set to 1) scores possible. Sequences with a score of at least 0.85 were considered to contain a motif match.

### Computational modeling of terminator strength
To predict terminator strength, we first build a lasso regression model using the glmnet package (version 4.1.7) in R to predict terminator strength based on the counts of all possible, overlapping 6-mer counts. The regression model was trained on 70% of the data and tested on the remaining 30%.

For our second model, we built a convolutional neural net model using EUGENe (version 0.0.6)[80] and PyTorch ([81]; version 1.11.0) in Python (version 3.8.10). We used a "DenseNet"[58] architecture adapted from iCREPCP[59]. The model takes one-hot encoded DNA as an input which is fed to a convolutional layer with 128 filters and a kernel size of 5. This layer is followed by four dense blocks consisting of 6, 12, 24, and 16 convolutional layers with 12 filters each and a kernel size of 3. In each dense block, the output of a convolutional layer is appended to its input and the combined output is used as the input for the subsequent layer. Between each dense block, the output feature map is reduced in size through convolution (using half as many filters as the input and a kernel size of 1) and average pooling (with a kernel size of 2). The output of the final dense block is fed into a fully connected layer with two outputs corresponding to the terminator strength in tobacco leaves and maize protoplasts, respectively. Terminator sequences from our original library that were detected in tobacco leaves and maize protoplast STARR-seq experiments ($n = 53,409$) were used for model training and evaluation. Of this data, 81% were used to train the model, 9% were used as a validation set during training and the remaining 10% were used to test the generalizability and accuracy of the trained model. Feature attributions for sequences in the test set were calculated using the DeepLIFT method[60] implemented in EUGENe. These feature attributions were used as input for TF-MoDISco-lite (version 2.0.6)[61] to extract motifs that increase or decrease terminator strength. Since we were looking for RNA motifs, we modified the TF-MoDISco-lite algorithm to not consider reverse complemented seqlets for clustering, alignments, or pattern generation. The code for model training and evaluation, and the trained model are available on GitHub (https://github.com/lampoona/Terminators-Plant-STARR-seq).

### In silico evolution of terminator sequences
We used the DenseNet model to iteratively improve terminator strength. We randomly selected 222 sequences (111 each from *Arabidopsis* and maize) from the test set of our model to ensure that the model had not seen these sequences during training. In each iteration,

we generated every possible point mutation for each sequence, scored them with the DenseNet model, and kept the sequence variant with the highest predicted strength in tobacco leaves, maize protoplasts, or both (using the sum of the activity in the individual systems) systems for the next iteration. We then experimentally determined the terminator strength of the sequences after three and ten rounds of evolution using Plant STARR-seq in both tobacco leaves and maize protoplasts.

## Dual-luciferase assay

For the dual-luciferase assay, terminators were cloned downstream of a nanoluciferase gene under control of the 35S enhancer and minimal promoter in the plasmid pDLterm (https://www.addgene.org/211903/). The plasmid also harbors a luciferase gene under control of the *Arabidopsis UBQ10* promoter. For the dual-luciferase assay in tobacco, two independent biological replicates, with two technical replicates each, were performed. The dual-luciferase reporter plasmids were introduced into *Agrobacterium tumefaciens* strain GV3101 (harboring the virulence plasmid pMP90 and the helper plasmid pSoup) by electroporation and used for transient transformation of tobacco leaves. Two days after the transformation, a total of 4 leaf discs from the third and fourth leaf of the tobacco plants was collected using a cork borer (4 mm diameter). The leaf discs were transferred to 1.5 mL tubes filled with approximately 10 glass beads (1 mm diameter), snap-frozen in liquid nitrogen, and disrupted by shaking twice for 5 sec in a Silamat S6 (Ivoclar) homogenizer. The leaf disc debris was resuspended in 100 μL 1X Passive Lysis Buffer (Promega). The solution was cleared by centrifugation (5 min, 20,000 x g, RT) and an aliquot of the supernatant was diluted 1:10 with 1X passive lysis buffer. Luciferase and nanoluciferase activities were measured on a Biotek Synergy H1 plate reader using the Promega Nano-Glo Dual-Luciferase Reporter Assay System according to the manufacturer's instructions. Specifically, 10 μL of the diluted leaf extracts were combined with 75 μL ONE-Glo EX Reagent, mixed for 3 min at 425 rpm, and incubated for 2 min before measuring luciferase activity. Subsequently, 75 μL NanoDLR Stop&Glo Reagent were added to the sample. After 3 min mixing at 425 rpm and 12 min incubation, nanoluciferase activity was measured.

For the dual-luciferase assay in maize protoplasts, two independent biological replicates, with a technical replicate for one of them, were performed. One million protoplasts were transformed with 15 μg of the dual-luciferase reporter plasmid following the same transformation protocol as used for Plant STARR-seq (see above). After incubation overnight in the dark, the transformed protoplasts were washed twice with 1 mL incubation solution and lysed by adding 100 μL 1X passive lysis buffer. The protoplast lysate was diluted 1:10 with 1X passive lysis buffer and used to measure luciferase and nanoluciferase activities with the same protocol as used for the tobacco leaf extracts. Raw values and calculated scores from assay are provided in Supplementary Data 5.

## 3′ end sequencing

We used 3′ end sequencing to determine the cleavage and polyadenylation site for each terminator. For cDNA synthesis, we incubated 5 μL of the extracted mRNA from the second Plant STARR-seq replicate in tobacco leaves with 5 μL RNase free water, 1 μL 10 mM DNTPs, and 2 μL 25 mM UMI-containing oligo-dT primer for 5 min at 55 °C. The solution was supplemented with 4 μL 5X SuperScript IV buffer, 1 μL 100 mM DTT, 1 μL RNaseOUT, and 1 μL Superscript IV reverse transcriptase. The reaction was incubated for 2 min each at 4 °C, 10 °C, 20 °C, 30 °C, 40 °C and 50 °C, and finally for 15 min at 55 °C. The sample was supplemented with 1 μL RNase H (NEB) and incubated for 30 min at 37 °C. The cDNA was purified using the Zymo Clean&Concentrate-5 kit and PCR-amplified using indexed sequencing primers specific to the end of the GFP construct and the adapter added using the RT primer (Fig. 8a). Terminators were sequenced using an Illumina

NextSeq 2000 platform with a 222-nucleotide read from the 5′ end of the amplicon. The paired read was used to sequence the 8-nucleotide UMI. UMIs and terminator reads were linked using UMI-Tools (version 1.1.2)[82]. Poly-A tails of each read were removed using CutAdapt (version 2.5)[83]. Terminators were aligned to the designed terminator library plus 52 nucleotide of plasmid backbone sequence using BowTie2 (version 2.4.1)[79] and reads that had a map quality of 0 or 1 were removed using Samtools (version 1.9)[84]. Duplicate terminator-UMI pairs were removed and terminators with fewer than 20 supporting reads were discarded. Bam files for each terminator were generated, and cleavage sites were determined using a custom R script available on GitHub (https://github.com/lampoona/Terminators-Plant-STARR-seq). Briefly, the algorithm identifies local maxima in the distribution of read lengths per terminator. Cleavage probability was calculated on a per terminator basis as the percentage of reads shorter than 171 nucleotides (Supplementary Data 1). Major cleavage and polyadenylation sites per terminator are reported in Supplementary Data 6.

## RT-PCR and gel electrophoresis

To visualize terminator cleavage patterns, RNA was extracted from tobacco leaves transiently transformed with dual-luciferase reporter constructs. A total of 8 leaf discs (4 mm diameter) were collected two days after transformation. The fresh leaf discs were transferred to a 2 mL tube containing a 5 mm stainless steel bead and 500 μL buffer RLT (QIAGEN). The leaf discs were disrupted using a TissueLyser LT (QIAGEN) bead mill for 3 min at 50 Hz. Debris was pelleted by centrifugation (2 mi, 20,000 x g, RT) and the supernatant was subjected to RNA extraction using the QIAGEN RNease Plant Mini Kit according to the manufacturer's instructions. Reverse transcription of the extracted RNA was performed in the same way as for the 3′ end sequencing. The resulting cDNA was PCR-amplified with primers binding to the 3′ end of the nanoluciferase gene and the 5′ end of the reverse transcription primer. The PCR products were separated by gel electrophoresis and visualized with SYBR green.

## Statistics and reproducibility

All STARR-seq experiments and nanoluciferase experiments were conducted with at least two biological replicates. Terminator strength is the average of the two replicates. Data were excluded from the experiment if not present in both replicates. Spearman and Pearson's correlation are reported for reproducibility. Significance was calculated using the two-sided Wilcoxon rank-sum test. The experiments were not blinded to the investigators. However, assessing element function in the context of a large library with barcode sequencing is by nature a randomized experiment.

## Reporting summary

Further information on research design is available in the Nature Portfolio Reporting Summary linked to this article.

## Data availability

All sequencing results are deposited in the NCBI Sequence Read Archive under the BioProject accession PRJNA991151. The processed data are provided in this paper in Supplementary Data 1, 2, 5 and 6.

## Code availability

The code used in this study is available on GitHub (https://github.com/lampoona/Terminators-Plant-STARR-seq).

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

## Acknowledgements

This work was supported by the National Science Foundation (RESEARCH-PGR grant no. 1748843 to E.S.B., S.F. and C.Q. and Plant-SynBio grant no. 2240888 to C.Q.), the German Research Foundation (DFG; postdoctoral fellowship no. 441540116 to T.J.), the National Institute of Health (T32 training grant no. HG000035 to J.T., NIGMS grant no. R01-GM079712 to J.T.C. and C.Q., and NIGMS MIRA grant no. 1R35GM139532), and the United States Department of Agriculture (NIFA postdoctoral fellowship no. 2023-67012-39445 to N.A.M.).

## Author contributions

All authors conceived and interpreted experiments; S.G., T.J., J.T., and N.A.M. performed experiments; S.G., T.J., and K.B. analyzed the data; S.G., T.J., and T.W. did the in silico modeling. S.G., T.J., C.Q., and S.F. prepared the manuscript and figures.

## Competing interests

The authors declare no competing interests.
