## [Peer Review File · Nature Communications]

Reviewers' comments:

Reviewer #1 (Remarks to the Author):

The manuscript presents a comprehensive analysis of terminator elements in plants using the Plant STARR-seq massively parallel reporter assay. This assay allows the sensitive measurement of the activity of cis-regulatory elements. In this study, the authors establish a definition of terminator strength that reflects both transcriptional activity and RNA stability and apply it to investigate over 50,000 terminators from *Arabidopsis thaliana* and *Zea mays*. One of the major issues in this article is the lack of clarity in the definition of “terminator” and “terminator strength”. The authors have not provided sufficient explanations to elucidate these concepts. To improve the clarity and comprehensibility of the manuscript, it is suggested that the authors provide a more detailed explanation in the article. This will assist readers in understanding the significance of this definition and its relevance to the study. Consequently, the current manuscript remains controversial in its present form. However, if the authors address these issues and revise the manuscript accordingly, its potential for publication could be reassessed.

1. In the introduction, the authors cite the study of (Kumar et al. 2019) and state that the 3' end of a gene — commonly called a terminator — contains sequence motifs necessary for mRNA 3' end maturation, cleavage, and polyadenylation. However, in the study cited by the authors (Kumar et al., 2019), no mention of the concept of terminators was identified. As a result, the authors' claim regarding the existence of sequence motifs in the 3' end of genes, commonly known as terminators, appears to lack supporting evidence. To strengthen this aspect of the manuscript, the authors need to conduct a more extensive literature review to identify additional studies that clearly define and characterize terminators in the context of mRNA 3' end maturation in plant systems.

2. The authors introduce the formula for terminator strength, which is described as the enrichment of barcodes in RNA relative to DNA. However, the scientific basis for this formula is not adequately explained, making it incomprehensible. To improve the clarity and comprehension of the manuscript, the authors need to provide a more detailed explanation of the rationale behind this calculation. This would assist readers in understanding the significance of this definition and its relevance to the study.

3. Based on 1 and 2, the current title does not align adequately with the content of the study. Furthermore, the title suffers from being excessively broad, as the study's observations are limited to just two plant species and may not be applicable to all plants universally. The authors need to select a more specific title that accurately reflects the precise scope and focus of their research.

4. The authors define the terminator as a 170-nucleotide sequence spanning from position -150 to +20 relative to the cleavage and polyadenylation sites. However, more evidence is needed to support the appropriateness of this terminator interval. Providing additional justification and supporting data would strengthen the validity of your findings.

5. The authors present data but fail to provide statistical analysis to support their conclusions, such as Figure 3b and 3c. To enhance the credibility and rigor of the study, it is imperative that the authors include appropriate statistical tests to validate their results and draw accurate inferences.

6. In Figure 4,5,6, the authors briefly mention the high overlap between the terminator sequence and the polyadenylation signal region in plants. To improve the clarity of the article, it is important to further expand upon the distinction between the polyadenylation signal and the terminator element. This will help readers understand the specific role and function of the terminator element in transcription termination.

7. On page 9, the authors claim that the newly generated terminators exhibit greater strength compared to the t35S terminator. However, the manuscript lacks experimental evidence to support these claims. Including additional experimental results to validate the strength of the newly generated terminators would enhance the credibility of your findings and broaden the practical applications of this research. This evidence would help evaluate the potential value of these terminators as tools in genetic engineering.

Reviewer #3 (Remarks to the Author):

The authors have adapted a genome-wide approach for identification of functional elements (STARR-seq) for the characterization of plant transcription terminators. For this, the authors synthesized a library of Arabidopsis and maize terminators guided by collections of experimentally-defined poly(A) sites from high-throughput poly(A) site profiling. These terminators (as well as some different sets of control sequences) were introduced into the STARR-seq pipeline for subsequent experimental manipulation and data processing. The end result was a compilation of relative terminator strengths, the latter metric reflecting the relative abundances of transcripts carrying the respective terminator sequences. The standard STARR-seq output was used to make inferences about terminator function. These include evidence for functionality of sequence motifs shown by others to be important for poly(A) site function, for a degree of species specificity when it comes to terminator and motif functioning, and for an ability to design efficient terminators using computational approaches trained on the results of the STARR-seq study.

The chief weakness of this study pertains to the matter of what the STARR-seq approach is actually measuring. The commonly-accepted (and, in this reviewer's opinion, entirely correct) measure of terminator efficiency is analogous to what the authors of this study describe as "cleavage probability". However, the terminator against which all other elements in this study are normalized – the so-called cauliflower mosaic virus polyadenylation site (t35S) – has a very low cleavage probability (if I am reading Supplementary Table 2 correctly, the value for this element is about 0.036), with almost all of the 3' sequencing reads for this element ending in downstream

sequences. In other experiments with this terminator (such as Mogen BD et al. *Plant Cell*. 1990 2:1261-72, Sanfaçon et al., *Genes Dev.* 1991 5:141-9, Mogen BD et al., *Mol Cell Biol.* 1992 12:5406-14, and Rothnie et al., *EMBO J.* 1994 13:2200-10.), there is essentially no readthrough (cleavage probabilities greater than 0.95) into downstream regions (that are different in the different studies). This is almost a 25-fold discrepancy. The results from the STARR-seq experiment are so different from these that the utility of STARR-seq to measure terminator functioning must be questioned. Specifically, it seems very likely that some aspect of the design of the reporter system is having an unduly large effect on poly(A) site functioning. This concern is magnified by the reported cleavage probabilities for other sets of elements – the numerous instances of cleavage probabilities for experimentally-determined poly(A) sites below 0.8 or even 0.6 raise serious questions about the claims regarding terminator function. (In this reviewer's opinion, this is the main reason why terminator efficiency does not correlate with differential usage of poly(A) sites in those gene in which primary and secondary sites were studied, as in Supplemental Fig. 8. The assay used here is not a reliable indicator of poly(A) site functionality or efficiency.)

Another indication that the reporter gene system is a confounding factor is shown in Fig. 9A. It is remarkable that so many random sequences (especially in *Arabidopsis*) can apparently function as poly(A) signals (based on their high cleavage probabilities). This makes no sense, and emphasizes that the STARR-seq approach here is not assaying terminator function or efficiency. Essentially, claims that terminator functioning is actually being measured by STARR-seq must be confirmed by independent means that do not involve computational or high throughput sequencing studies.

The question arises as to what is actually being measured? Besides terminator function, formal possibilities include transcription and RNA stability. The results reported here do not definitively resolve this issue, but other reports may hold clues. In particular, Pauli et al. (*J Virol.* 2004 78:12120-8.) have shown that the beginning of the CaMV 35S RNA possesses enhancer activities. It is conceivable that the t35S element, that is embedded within the beginning of the 35S RNA, may also possess enhancer elements. Other studies cited by the authors in other papers (Ricci et al. *Nat. Plants* 2019 5: 1237–1249. and Sun et al., *Genomics Proteomics Bioinformatics* 2019 17: 140–153.) have shown that enhancer elements placed in the 3' regions of the STARR system can function. Given that the t35S is such a poor terminator (judging from the cleavage probability) but yields very high reporter expression levels, it seems likely that what is being called terminator efficiency here is actually some measure of enhancer activity. Associations of motifs that resemble previously-characterized polyadenylation-related elements (such as AAUAAA and UGUA) with enhanced expression probably reflects overlaps between the set of motifs recognized by the totality of plant transcription factors and these particular sequences, and also the obvious biases associated with the ways the authors created the set of elements to be tested.

Other concerns and discussion that the authors should consider:

1. The authors assembled their set of Arabidopsis terminators in part using results of a study (Thomas et al., 2012) of poly(A) site choice in a mutant deficient in functioning of a component of the polyadenylation machinery. Did the authors include the set of sites reported in Thomas et al. as occurring only in the mutant? If so, this is a confounding factor that may introduce a degree of “noise” in the results and conclusions. (There is opportunity here – perhaps the mutant-specific terminators have different behaviors in the STARR-seq system.)

2. A systematic analysis of the effects of point mutations on AAUAAA function in plants has been conducted (Rothnie et al., EMBO J. 1994 13:2200-10). The results in Supplemental Fig 7 should be compared with this study. (Of course, discrepancies likely reflect the unsuitability of the STARR-seq approach to study terminator function.)

3. Throughout this study the motif UGUA is mentioned and its functionality measured. However, nowhere are there citations of the pioneering studies from the Hohn laboratory that showed the importance of this motif in plants. This is not merely a criticism of a lack of citation. These classical experiments have data that are relevant to the STARR-seq study.

4. The authors should include in their discussions the observation that AU-rich sequences themselves can confer poly(A) signal-like function in maize (see Luehrsen and Walbot, Genes Dev. 1994 8:1117-30).

5. Aside from one mention in the introduction, the authors do not discuss results that may shed light into the composition and functioning of the so-called Cleavage Element.

6. Legends and descriptions for the Supplemental Tables need to be provided. It is very difficult to follow what is being presented in some of these.

7. Along these lines, the mined data used to create the plots shown in Supplemental Fig. 3 should be provided in a Supplemental Table.

Reviewer #1 (Remarks to the Author):

The manuscript presents a comprehensive analysis of terminator elements in plants using the Plant STARR-seq massively parallel reporter assay. This assay allows the sensitive measurement of the activity of cis-regulatory elements. In this study, the authors establish a definition of terminator strength that reflects both transcriptional activity and RNA stability and apply it to investigate over 50,000 terminators from Arabidopsis thaliana and Zea mays. One of the major issues in this article is the lack of clarity in the definition of “terminator” and “terminator strength”. The authors have not provided sufficient explanations to elucidate these concepts. To improve the clarity and comprehensibility of the manuscript, it is suggested that the authors provide a more detailed explanation in the article. This will assist readers in understanding the significance of this definition and its relevance to the study. Consequently, the current manuscript remains controversial in its present form. However, if the authors address these issues and revise the manuscript accordingly, its potential for publication could be reassessed.

We thank the reviewer for their positive assessment of our manuscript and we will gladly address their comments to improve our manuscript. As requested, we will include a detailed and well-justified definition of plant terminators, as described below.

1. In the introduction, the authors cite the study of (Kumar et al. 2019) and state that the 3' end of a gene — commonly called a terminator — contains sequence motifs necessary for mRNA 3' end maturation, cleavage, and polyadenylation. However, in the study cited by the authors (Kumar et al., 2019), no mention of the concept of terminators was identified. As a result, the authors' claim regarding the existence of sequence motifs in the 3' end of genes, commonly known as terminators, appears to lack supporting evidence. To strengthen this aspect of the manuscript, the authors need to conduct a more extensive literature review to identify additional studies that clearly define and characterize terminators in the context of mRNA 3' end maturation in plant systems.

We thank the reviewer for identifying this critical weakness in the original manuscript. We cited *Kumar et. al 2019* because this paper provides a careful description of nascent RNA cleavage and polyadenylation. We should have provided citations that define plant terminators and their role in cleavage, polyadenylation, and transgene expression. As reviewer 3 points out, we have been amiss by not citing foundational papers in the field. In the revised manuscript, we are amending this oversight by citing important earlier studies throughout the revised manuscript (e.g., Rothnie 1996; Rothnie et al. 1994; Sanfacon et al. 1991; Hunt 2008).

Terminator definition: Our usage of the term ‘terminator’ follows Felippes et. al 2022 that described terminators as “the genic elements located downstream of a coding sequence that, once transcribed, is recognized by different protein complexes responsible for...cleavage and polyadenylation of the nascent pre-mRNA transcript...for mRNA biogenesis.”

In the revised manuscript, we state clearly that our assay determines the extent to which 3' sequence surrounding experimentally determined polyadenylation and cleavage sites affect transcript levels. These sequences, termed terminators henceforth, may affect mRNA expression, processing, and stability.

The revised manuscript states on **page 1, line 39**: “For example, the 3’ end of a gene contains sequence motifs necessary for mRNA 3’ end maturation, cleavage, and polyadenylation (Kumar et al. 2019; A. G. Hunt 1994; Arthur G. Hunt 2011; de Felippes and Waterhouse 2023; Loke et al. 2005; B. Tian and Graber 2012, Rothnie et al. 1994; Rothnie 1996). In keeping with established nomenclature (Brooks et al. 2023; de Felippes and Waterhouse 2023; C. Tian et al. 2022; Rothnie 1996), throughout this manuscript, we refer to sequences surrounding a cleavage and polyadenylation site as terminators.”

2. The authors introduce the formula for terminator strength, which is described as the enrichment of barcodes in RNA relative to DNA. However, the scientific basis for this formula is not adequately explained, making it incomprehensible. To improve the clarity and comprehension of the manuscript, the authors need to provide a more detailed explanation of the rationale behind this calculation. This would assist readers in understanding the significance of this definition and its relevance to the study.

Terminators affect gene expression (de Felippes and Waterhouse 2023; Bernardes and Menossi 2020). Therefore, we use reporter gene expression as a proxy for how efficiently a terminator is cleaved and polyadenylated. To quantify reporter gene expression, we use the log fold ratio of the number of RNA molecules produced normalized by the number of DNA molecules per terminator (as described in Jores et al. 2020).

More specifically, the reads for each unique barcode associated with a terminator sequence are counted in the DNA input and cDNA (derived from RNA) output samples. Barcode counts below 5 are discarded. The output enrichment of a terminator is calculated as the median enrichment of all unique barcodes linked to it. We calculated terminator strength as the log₂ fold of the terminator enrichment normalized to the enrichment of the CaMV 35 terminator (t35S), a strong viral terminator (see reference above). The average terminator strength of the biological replicates is reported.

In the revised manuscript, we now explain the scientific basis for this approach to make it more accessible to readers. **Page 3, line 130**: “Strong terminators yield higher transcript levels due to improved 3’ end processing or transcript stability as compared to weak terminators (Felippes 2023; Bernardes 2020). The number of transcripts (i.e. RNA reads) per DNA template (i.e. DNA reads) is therefore a direct measure of the strength of a terminator.”

3. Based on 1 and 2, the current title does not align adequately with the content of the study. Furthermore, the title suffers from being excessively broad, as the study's observations are limited to just two plant species and may not be applicable to all plants universally. The authors need to select a more specific title that accurately reflects the precise scope and focus of their research.

We changed the title to: “Arabidopsis and Maize Terminator Strength is Determined by GC Content, Polyadenylation Motifs and Cleavage Probability”

4. The authors define the terminator as a 170-nucleotide sequence spanning from position -150 to +20 relative to the cleavage and polyadenylation sites. However, more evidence is needed to support the appropriateness of this terminator interval. Providing additional justification and supporting data would strengthen the validity of your findings.

This is an important point. The length of the tested terminator sequences was determined by considering known terminator biology and the technical limitations of array synthesis. In the revised manuscript, we explain our rationale for choosing the -150 to +20 window around experimentally determined cleavage sites as terminators on **page 3, line 100**: “The average 3’ untranslated region (UTR) length in Arabidopsis and maize is 242 bp and 310 bp, respectively (Srivastava 2018; Jafar 2019). However, due to technical limitations in DNA array-synthesis, we were limited to sequences of 170 nucleotides. Previous studies on plant terminators revealed that most elements required for efficient mRNA 3’ end processing reside within approximately 150 bp upstream of the cleavage and polyadenylation site (Loke et al. 2005; A. G. Hunt 1994; Shen et al. 2008; Bernardes and Menossi 2020; de Felippes and Waterhouse 2023, Rothnie 1996). Furthermore, studies in yeast and animals revealed that sequence elements downstream of the cleavage site can also affect polyadenylation (B. Tian and Graber 2012; Bernardes and Menossi 2020; de Felippes and Waterhouse 2023). Although such elements have not been reported in plants, we included the 20 nucleotides downstream of the cleavage site in our candidate sequences to test if they have an effect of terminator activity. For these reasons, we defined a terminator as the 170 nucleotide sequence from position -150 to +20 relative to a cleavage and polyadenylation site (position 0) in this study.”

5. The authors present data but fail to provide statistical analysis to support their conclusions, such as Figure 3b and 3c. To enhance the credibility and rigor of the study, it is imperative that the authors include appropriate statistical tests to validate their results and draw accurate inferences.

We agree. We added the results of statistical analyses to **Figures 3b and c, 4d–g, 7b and c, 8g, 9a, b, and d, and Supplementary Figures 8 and 9.**

6. In Figure 4,5,6, the authors briefly mention the high overlap between the terminator sequence and the polyadenylation signal region in plants. To improve the clarity of the article, it is important to further expand upon the distinction between the polyadenylation signal and the terminator element. This will help readers understand the specific role and function of the terminator element in transcription termination.

Our study focuses on effects of terminators on mRNA transcript levels, polyadenylation, and cleavage rather than transcription termination. However, these biological processes are linked, and we discuss this relationship in a new paragraph in the revised introduction (**Page 2, line 50**): “Messenger RNA cleavage and polyadenylation are tightly linked to and initiate transcription termination (Connelly and Manley 1988; Birse et al. 1998; Proudfoot 2011; Whitelaw and Proudfoot 1986); however, termination occurs up to 1 kb downstream of the mRNA cleavage site (Mo et al. 2021). The current model of how mRNA 3’ end processing leads to transcription termination combines features of two previously proposed models (Rosonina, Kaneko, and Manley 2006; Luo, Johnson, and Bentley 2006; Eaton and West 2020; Krzyszton et al. 2018): After mRNA cleavage, conformational changes of RNA polymerase II allosterically reduce its elongation rate (allosteric model). At the same time, the cleavage site enables a 5’ to 3’ exonuclease (Rat1 in yeast, XRN2 in humans, and XRN3 in *Arabidopsis*) to degrade the downstream RNA and initiate transcription termination when it catches up with the RNA polymerase (torpedo model). The exact site of transcription termination can further be affected

by RNA polymerase II pause sites, R loops formed between DNA and the nascent RNA, and DNA-binding proteins (Eaton and West 2020; Yu, Martin, and Michaels 2019).”

7. On page 9, the authors claim that the newly generated terminators exhibit greater strength compared to the t35S terminator. However, the manuscript lacks experimental evidence to support these claims. Including additional experimental results to validate the strength of the newly generated terminators would enhance the credibility of your findings and broaden the practical applications of this research. This evidence would help evaluate the potential value of these terminators as tools in genetic engineering.

The strength of the *in silico* evolved plant terminators was determined experimentally with Plant STARR-seq and the results are presented in **Figure 7b and c**.

As an additional validation, we cloned several *in silico* evolved terminators into dual-luciferase constructs and measured their strength in transiently transformed tobacco leaves. The results from these experiments are shown in **Figure 7d and e** and are discussed in the revised manuscript on **page 9 and line 396**: “We tested selected evolved terminators with the dual-luciferase assay and found strong correspondence with the Plant STARR-seq results (Fig. 7d, e)”.

Reviewer #3 (Remarks to the Author):

The authors have adapted a genome-wide approach for identification of functional elements (STARR-seq) for the characterization of plant transcription terminators. For this, the authors synthesized a library of Arabidopsis and maize terminators guided by collections of experimentally-defined poly(A) sites from high-throughput poly(A) site profiling. These terminators (as well as some different sets of control sequences) were introduced into the STARR-seq pipeline for subsequent experimental manipulation and data processing. The end result was a compilation of relative terminator strengths, the latter metric reflecting the relative abundances of transcripts carrying the respective terminator sequences. The standard STARR-seq output was used to make inferences about terminator function. These include evidence for functionality of sequence motifs shown by others to be important for poly(A) site function, for a degree of species specificity when it comes to terminator and motif functioning, and for an ability to design efficient terminators using computational approaches trained on the results of the STARR-seq study.

We thank the reviewer for their careful and accurate summary of our study.

The chief weakness of this study pertains to the matter of what the STARR-seq approach is actually measuring. The commonly-accepted (and, in this reviewer’s opinion, entirely correct) measure of terminator efficiency is analogous to what the authors of this study describe as “cleavage probability”. However, the terminator against which all other elements in this study are normalized – the so-called cauliflower mosaic virus polyadenylation site (t35S) – has a very low cleavage probability (if I am reading Supplementary Table 2 correctly, the value for this element is about 0.036), with almost all of the 3’ sequencing reads for this element ending in downstream sequences. In other experiments with this terminator (such as Mogen BD et al. Plant Cell. 1990 2:1261-72, Sanfaçon et al., Genes Dev. 1991 5:141-9, Mogen BD et al., Mol Cell Biol. 1992 12:5406-14, and Rothnie et al., EMBO J. 1994 13:2200-10.), there is essentially

no readthrough (cleavage probabilities greater than 0.95) into downstream regions (that are different in the different studies). This is almost a 25-fold discrepancy. The results from the STARR-seq experiment are so different from these that the utility of STARR-seq to measure terminator functioning must be questioned. Specifically, it seems very likely that some aspect of the design of the reporter system is having a unduly large effect on poly(A) site functioning.

We thank the reviewer for catching our error in calculating the cleavage probability of the 35S terminator. This error was caused by a bug in our code: we considered cuts within the 170 bp ‘terminator’ sequence relative to cuts downstream. All array-synthesized candidate terminator sequences are 170 bp long; however, the four viral and bacterial control sequences, including the 35S terminator, are longer. Their increased length was not taken into account in our original code and analysis. After fixing the code, the cleavage probability for the 35S terminator is 0.99, consistent with the studies mentioned by the reviewer. To support the accuracy of the corrected result, we now include a plot showing our experimentally determined cleavage sites for the 35S terminator in the new **Supplementary Figure 10b**. We also discuss the concordance between our results and previous studies in the revised results section (**Page 9, line 422**): “Consistent with prior reports (Rothnie, Reid, and Hohn 1994; Mogen et al. 1992; Sanfaçon, Brodmann, and Hohn 1991; Mogen et al. 1990) we observed near complete cleavage of the 35S terminator with one major cleavage site at the same position as previously shown (Supplementary Fig. 10b).”

We apologize for the confusion that this error caused, and we are grateful for the opportunity to correct it. Please note that this error affected only the four control sequences; the results reported for all other terminators are accurate as reported in the original manuscript.

All major cleavage and polyadenylation sites derived from 3’ end sequencing are now reported in Supplementary Table 8 and included in the text (**page 18, line 817**).

This concern is magnified by the reported cleavage probabilities for other sets of elements – the numerous instances of cleavage probabilities for experimentally-determined poly(A) sites below 0.8 or even 0.6 raise serious questions about the claims regarding terminator function. (In this reviewer’s opinion, this is the main reason why terminator efficiency does not correlate with differential usage of poly(A) sites in those gene in which primary and secondary sites were studied, as in Supplemental Fig. 8. The assay used here is not a reliable indicator of poly(A) site functionality or efficiency.)

The vast majority (84%) of *Arabidopsis* terminators has a cleavage probability higher than 0.6. Even at the more stringent cut-off of 0.8, 73% of the *Arabidopsis* terminators pass this threshold. However, as pointed out by the reviewer, there are *Arabidopsis* terminators that show low cleavage probability. This result is neither surprising nor unexpected because ~70% of the genes in the *Arabidopsis* genome use more than one polyadenylation site (Wu et al. 2011). In other words, some level of polymerase read-through is expected for the majority of *Arabidopsis* terminators. We are confident that our assay accurately measures terminator activity because our results are consistent with prior knowledge: Cleavage probability is correlated with the presence of known polyadenylation motifs, terminator strength is positively correlated with cleavage probability, and controls such as the 35S terminator behave as previously described in studies using orthogonal methods. In the revised manuscript, we also provide results using an orthogonal method for terminator activity to provide further support (**Supplementary Fig. 3; Fig. 7d, e; Supplementary Fig. 10c; page 18, line 819; page 18, line 849**).

The raw and normalized data from the dual luciferase assays are now included as new Supplementary Table 7 and included in the text (**page 18, line 847**).

Another indication that the reporter gene system is a confounding factor is shown in Fig. 9A. It is remarkable that so many random sequences (especially in Arabidopsis) can apparently function as poly(A) signals (based on their high cleavage probabilities). This makes no sense, and emphasizes that the STARR-seq approach here is not assaying terminator function or efficiency.

The included 'random' sequences do not represent controls for which zero or even low cleavage probability is expected. These sequences were designed to test which nucleotide composition predisposes a sequence to function as a terminator. The sequences that show high cleavage probabilities are not random with regard to their nucleotide composition – they show high AU content. As the reviewer points out below in their critique point 4 “AU-rich sequences themselves can confer poly(A) signal-like function” (Luehrsen and Walbot 1994). This prior finding is consistent with our observation that high AU content is associated with high cleavage probability and explains why many of the 'random' *Arabidopsis* sequences, which are AU-rich, show high cleavage probability. Please note that the more GC-rich 'random' maize sequences display a lower cleavage probability on average, supporting this interpretation.

Essentially, claims that terminator functioning is actually being measured by STARR-seq must be confirmed by independent means that do not involve computational or high throughput sequencing studies.

To validate our findings with a non-sequencing-based approach, we tested the strength of a selected set of terminators using a dual-luciferase assay in transiently transformed leaves. The results from these experiments are shown in **Figure 7 and Supplementary Figure 3** and are discussed in the revised manuscript **on page 4, lines 158** “We sought to confirm our measure of terminator strength by Plant STARR-seq with an orthogonal assay. To do so, we conducted a dual-luciferase assay in both tobacco leaves and maize protoplasts for several weak, intermediate, and strong terminators. We observed a strong correlation between terminator strength measured by the two assays (Supplemental Fig. 3). Strong terminators yielded higher nanoluciferase activity than weak terminators”.

For the validation of evolved terminators, please see **page 9, lines 396 and Fig. 7d, e**. “We tested selected evolved terminators with the dual-luciferase assay and found strong correspondence with the Plant STARR-seq results (Fig. 7d, e)”.

We also confirmed the cleavage pattern of several terminators via RT-PCR and gel electrophoresis and the results are shown in **Supplementary Fig. 10** and summarized in the revised manuscript **on page 9, line 425**: “We confirmed the cleavage pattern of several terminators with RT-PCR and gel electrophoresis (Supplementary Fig. 10c)”.

The question arises as to what is actually being measured? Besides terminator function, formal possibilities include transcription and RNA stability. The results reported here do not definitively resolve this issue, but other reports may hold clues. In particular, Pauli et al. (J Virol. 2004 78:12120-8.) have shown that the beginning of the CaMV 35S RNA possesses enhancer

activities. It is conceivable that the t35S element, that is embedded within the beginning of the 35S RNA, may also possess enhancer elements. Other studies cited by the authors in other papers (Ricci et al. Nat. Plants 2019 5: 1237–1249. and Sun et al., Genomics Proteomics Bioinformatics 2019 17: 140–153.) have shown that enhancer elements placed in the 3' regions of the STARR system can function. Given that the t35S is such a poor terminator (judging from the cleavage probability) but yields very high reporter expression levels, it seems likely that what is being called terminator efficiency here is actually some measure of enhancer activity. Associations of motifs that resemble previously-characterized polyadenylation-related elements (such as AAUAAA and UGUA) with enhanced expression probably reflects overlaps between the set of motifs recognized by the totality of plant transcription factors and these particular sequences, and also the obvious biases associated with the ways the authors created the set of elements to be tested.

The reviewer correctly points out that the terminator strength measured in our study could be confounded by enhancer activity. However, for the following reasons, this is highly unlikely:

1. As we have shown previously, in our assay, enhancers do not affect transcription when located in the 3' UTR (Jores et al. 2020). Ricci et al. and Sun et al. used a different reporter construct and analysis approach for their studies.
2. Our *de novo* motif discovery yields strong matches with known polyadenylation motifs, but not with transcription factor binding sites.
3. We tested if terminator strength is correlated with the presence of a given transcription factor binding site. This approach was adapted from our previous study on core promoters, where we used it to successfully identify several transcription factor binding sites that are associated with increased promoter strength (Jores et al. 2021). However, when we applied this approach to our terminator data, we failed to detect meaningful associations between terminator strength and the presence of transcription factor binding sites. The sole exception were the AT-rich binding sites of HD-ZIP family transcription factors that resemble the known AAUAAA polyadenylation motif. While we cannot exclude that HD-ZIP transcription factors have an influence on our terminator strength measure, we consider this observation a likely artifact caused by the similarity of the HD-ZIP binding sites to the AAUAAA motif. The motif discovered *de novo* in our study matches the AAUAAA motif more closely than the HD-ZIP binding sites.

Other concerns and discussion that the authors should consider:

1. *The authors assembled their set of Arabidopsis terminators in part using results of a study (Thomas et al., 2012) of poly(A) site choice in a mutant deficient in functioning of a component of the polyadenylation machinery. Did the authors include the set of sites reported in Thomas et al. as occurring only in the mutant? If so, this is a confounding factor that may introduce a degree of “noise” in the results and conclusions. (There is opportunity here – perhaps the mutant-specific terminators have different behaviors in the STARR-seq system.)*

We only used the cleavage sites detected in wild-type plants. This clarification is now stated in the revised manuscript (**page 12, line 555**).

2. A systematic analysis of the effects of point mutations on AAUAAA function in plants has been conducted (Rothnie *et al.*, *EMBO J.* 1994 13:2200-10). The results in Supplemental Fig 7 should be compared with this study. (Of course, discrepancies likely reflect the unsuitability of the STARR-seq approach to study terminator function.)

We compared our results to those of Rothnie *et al.* (Rothnie, Reid, and Hohn 1994) and found a moderate correlation. A perfect correlation is not expected. Our study assessed naturally existing AAUAAA variants in the context of thousands of terminator sequences while the Rothnie *et al.* assessed this motif and its variants in a single sequence context. Natural sequences may contain compensatory variants. Please see the respective figure below.

Fig 1. Point-by-point response. Correlation between a mutational analysis of the AAUAAA motif of the 35S terminator and all possible motif single nucleotide variants in all tested maize and *Arabidopsis* terminators. (a) reproduction of AAUAAA motif data from Rothnie *et al.* (b) correlation plot. Each dot encompasses at least 2559 to 8075 terminator sequences measured in STARR-seq.

3. Throughout this study the motif UGUA is mentioned and its functionality measured. However, nowhere are there citations of the pioneering studies from the Hohn laboratory that showed the importance of this motif in plants. This is not merely a criticism of a lack of citation. These classical experiments have data that are relevant to the STARR-seq study.

We apologize for this oversight and cite the paper in question (Rothnie, Reid, and Hohn 1994) at the appropriate places in the revised manuscript. In addition to showing the importance of the UGUA motif, Rothnie *et al.* also found that this motif increased polyadenylation efficiency in an additive manner. Consistent with Rothnie *et al.*, we observe that the number of UGUA motifs in a given terminator correlates with its terminator strength and cleavage probability. We present this finding in **Supplementary Fig. 8e, f** and summarize it in the revised manuscript on **page 6, line 280**: “Consistent with prior reports (Rothnie, Reid, and Hohn 1994), the number of UGUA motifs also correlates with terminator strength and cleavage probability (Supplementary Fig. 8e, f).”

Furthermore, in the *in silico* evolution experiment, we observed that the model favored the insertion of multiple UGUA motifs into the evolved, stronger terminators. These results are shown in **Supplementary Figure 8g** and are discussed in the revised manuscript on **page 9, line 398**.

4. The authors should include in their discussions the observation that AU-rich sequences themselves can confer poly(A) signal-like function in maize (see Luehrsen and Walbot, Genes Dev. 1994 8:1117-30).

We included this observation in the revised manuscript (**Page 10, line 469**): “Our observation that AU-rich sequences show high cleavage probability is consistent with previous reports that the insertion of AU-rich sequences into transcripts can lead to their cleavage and polyadenylation in maize (Luehrsen and Walbot 1994).”

5. Aside from one mention in the introduction, the authors do not discuss results that may shed light into the composition and functioning of the so-called Cleavage Element.

We found a canonical cleavage element with a YA core sequence at our experimentally determined cleavage sites and discuss this in the revised text (**page 9, line 419**): “As expected, the vast majority of the cleavage and polyadenylation sites identified by this approach coincided with a CA or UA dinucleotide (Fig. 8b) matching the known cleavage element (A. G. Hunt 1994; Loke et al. 2005).”

Unfortunately, the cleavage element is too short and has too little information content to be identified by our *de novo* motif discovery pipeline or to be used in our motif correlation analysis.

6. Legends and descriptions for the Supplemental Tables need to be provided. It is very difficult to follow what is being presented in some of these.

We added legends and descriptions for all tables and supplementary tables in the new supplementary tables excel file. This file contains all supplementary tables as sheets.

7. Along these lines, the mined data used to create the plots shown in Supplemental Fig. 3 should be provided in a Supplemental Table.

Since we did not generate the data that we compare to terminator strength in Supplemental Fig. 3, we prefer not to include it in a Supplemental Table. Inclusion could lead some readers to believe erroneously that we generated this data. Instead, we provide citations to the original studies in the legend to Supplemental Fig. 3. The data is freely available from these sources.

References:

- Bernardes, Willian Souza, and Marcelo Menossi. 2020. “Plant 3’ Regulatory Regions From mRNA-Encoding Genes and Their Uses to Modulate Expression.” *Frontiers in Plant Science*. <https://doi.org/10.3389/fpls.2020.01252>.
- Birse, C. E., L. Minvielle-Sebastia, B. A. Lee, W. Keller, and N. J. Proudfoot. 1998. “Coupling Termination of Transcription to Messenger RNA Maturation in Yeast.” *Science* 280 (5361):

298–301.

- Brooks, Emily G., Estefania Elorriaga, Yang Liu, James R. Dudit, Guoliang Yuan, Chung-Jui Tsai, Gerald A. Tuskan, Thomas G. Ranney, Xiaohan Yang, and Wusheng Liu. 2023. "Plant Promoters and Terminators for High-Precision Bioengineering." *Biodesign Research* 5 (July): 0013.
- Connelly, S., and J. L. Manley. 1988. "A Functional mRNA Polyadenylation Signal Is Required for Transcription Termination by RNA Polymerase II." *Genes & Development* 2 (4): 440–52.
- Eaton, Joshua D., and Steven West. 2020. "Termination of Transcription by RNA Polymerase II: BOOM!" *Trends in Genetics: TIG* 36 (9): 664–75.
- Felippes, Felipe F. de, and Peter M. Waterhouse. 2023. "Plant Terminators: The Unsung Heroes of Gene Expression." *Journal of Experimental Botany* 74 (7): 2239–50.
- Hunt, A. G. 1994. "Messenger RNA 3' end Formation in Plants." *Annual Review of Plant Biology*.
- Hunt, Arthur G. 2011. "RNA Regulatory Elements and Polyadenylation in Plants." *Frontiers in Plant Science* 2: 109.
- Jores, Tobias, Jackson Tonnie, Michael W. Dorrity, Josh T. Cuperus, Stanley Fields, and Christine Queitsch. 2020. "Identification of Plant Enhancers and Their Constituent Elements by STARR-Seq in Tobacco Leaves." *The Plant Cell* 32 (7): 2120–31.
- Jores, Tobias, Jackson Tonnie, Travis Wrightsman, Edward S. Buckler, Josh T. Cuperus, Stanley Fields, and Christine Queitsch. 2021. "Synthetic Promoter Designs Enabled by a Comprehensive Analysis of Plant Core Promoters." *Nature Plants* 7 (6): 842–55.
- Krzyszton, Michal, Monika Zakrzewska-Placzek, Aleksandra Kwasnik, Norbert Dojer, Wojciech Karlowski, and Joanna Kufel. 2018. "Defective XRN3-Mediated Transcription Termination in Arabidopsis Affects the Expression of Protein-Coding Genes." *The Plant Journal: For Cell and Molecular Biology* 93 (6): 1017–31.
- Kumar, Ananthanarayanan, Marcello Clerici, Lena M. Muckenfuss, Lori A. Passmore, and Martin Jinek. 2019. "Mechanistic Insights into mRNA 3'-End Processing." *Current Opinion in Structural Biology* 59 (December): 143–50.
- Loke, Johnny C., Eric A. Stahlberg, David G. Strenski, Brian J. Haas, Paul Chris Wood, and Qingshun Quinn Li. 2005. "Compilation of mRNA Polyadenylation Signals in Arabidopsis Revealed a New Signal Element and Potential Secondary Structures." *Plant Physiology* 138 (3): 1457–68.
- Luehrsen, K. R., and V. Walbot. 1994. "Intron Creation and Polyadenylation in Maize Are Directed by AU-Rich RNA." *Genes & Development* 8 (9): 1117–30.
- Luo, Weifei, Arlen W. Johnson, and David L. Bentley. 2006. "The Role of Rat1 in Coupling mRNA 3'-End Processing to Transcription Termination: Implications for a Unified Allosteric-Torpedo Model." *Genes & Development* 20 (8): 954–65.
- Mogen, B. D., M. H. MacDonald, R. Graybosch, and A. G. Hunt. 1990. "Upstream Sequences Other than AAUAAA Are Required for Efficient Messenger RNA 3'-End Formation in Plants." *The Plant Cell* 2 (12): 1261–72.
- Mogen, B. D., M. H. MacDonald, G. Leggewie, and A. G. Hunt. 1992. "Several Distinct Types of Sequence Elements Are Required for Efficient mRNA 3' End Formation in a Pea rbcS Gene." *Molecular and Cellular Biology* 12 (12): 5406–14.
- Mo, Weipeng, Bo Liu, Hong Zhang, Xianhao Jin, Dongdong Lu, Yiming Yu, Yuelin Liu, et al. 2021. "Landscape of Transcription Termination in Arabidopsis Revealed by Single-Molecule Nascent RNA Sequencing." *Genome Biology* 22 (1): 322.
- Proudfoot, Nick J. 2011. "Ending the Message: poly(A) Signals Then and Now." *Genes & Development* 25 (17): 1770–82.
- Rosonina, Emanuel, Syuzo Kaneko, and James L. Manley. 2006. "Terminating the Transcript: Breaking up Is Hard to Do." *Genes & Development* 20 (9): 1050–56.
- Rothnie, H. M., J. Reid, and T. Hohn. 1994. "The Contribution of AAUAAA and the Upstream Element UUUGUA to the Efficiency of mRNA 3'-End Formation in Plants." *The EMBO*

Journal 13 (9): 2200–2210.

- Sanfaçon, H., P. Brodmann, and T. Hohn. 1991. "A Dissection of the Cauliflower Mosaic Virus Polyadenylation Signal." *Genes & Development* 5 (1): 141–49.
- Shen, Yingjia, Guoli Ji, Brian J. Haas, Xiaohui Wu, Jianting Zheng, Greg J. Reese, and Qingshun Quinn Li. 2008. "Genome Level Analysis of Rice mRNA 3'-End Processing Signals and Alternative Polyadenylation." *Nucleic Acids Research* 36 (9): 3150–61.
- Tian, Bin, and Joel H. Graber. 2012. "Signals for Pre-mRNA Cleavage and Polyadenylation." *Wiley Interdisciplinary Reviews. RNA* 3 (3): 385–96.
- Tian, Chenfei, Yixin Zhang, Jianhua Li, and Yong Wang. 2022. "Benchmarking Intrinsic Promoters and Terminators for Plant Synthetic Biology Research." *BioDesign Research* 2022. <https://downloads.spj.sciencemag.org/bdr/2022/9834989.pdf>.
- Whitelaw, E., and N. Proudfoot. 1986. "Alpha-Thalassaemia Caused by a poly(A) Site Mutation Reveals That Transcriptional Termination Is Linked to 3' End Processing in the Human Alpha 2 Globin Gene." *The EMBO Journal* 5 (11): 2915–22.
- Wu, Xiaohui, Man Liu, Bruce Downie, Chun Liang, Guoli Ji, Qingshun Q. Li, and Arthur G. Hunt. 2011. "Genome-Wide Landscape of Polyadenylation in Arabidopsis Provides Evidence for Extensive Alternative Polyadenylation." *Proceedings of the National Academy of Sciences of the United States of America* 108 (30): 12533–38.
- Yu, Xuhong, Pascal G. P. Martin, and Scott D. Michaels. 2019. "BORDER Proteins Protect Expression of Neighboring Genes by Promoting 3' Pol II Pausing in Plants." *Nature Communications* 10 (1): 4359.

REVIEWER COMMENTS

Reviewer #1 (Remarks to the Author):

The authors' study delves into the influence of 3' UTR sequences on gene expression, with a focus on their composition, GC content, and motifs, making a significant contribution to our understanding of post-transcriptional regulation. Despite addressing some concerns raised during the peer review, the revised manuscript still harbors critical issues that warrant further attention. My primary concern revolves around the use of STARR-seq to evaluate "Terminator strength." In particular, I question the validity of using RNA-to-DNA ratios as a metric. Although this ratio is a reliable indicator of promoter activity—where a stronger promoter results in higher RNA levels—this logic may not extend to 3' UTR sequences functioning as termination signals. Contrary to expectations, higher RNA production does not necessarily imply more efficient termination. In fact, one could argue that effective termination might lead to reduced transcript levels downstream.

1. Regarding the response to question 2, "Therefore, we use reporter gene expression as a proxy for the efficiency with which a terminator is cleaved and polyadenylated," employing gene expression as a surrogate for terminator efficiency remains a matter of debate. de Felippes and Waterhouse (2023) discussed how poly(A) tail length affects gene expression, and Bernardes and Menossi (2020) examined the role of 3' Regulatory Regions in gene expression. Nevertheless, these studies fall short of providing concrete evidence that transcript levels can reliably estimate terminator efficiency.

2. In response to question 6, "Our study primarily investigates the impact of terminators on mRNA transcript levels, polyadenylation, and cleavage, rather than on transcription termination itself," it appears the essence of the study is to examine the effects of 3' UTR sequences on gene expression, not to directly assess their termination efficiency. It would be beneficial to make this distinction clearer in the manuscript to ensure readers fully understand the innovative approach and its significance for elucidating the role of 3' UTRs in gene regulation.

Reviewer #3 (Remarks to the Author):

This manuscript is a revision of an earlier one (previously entitled "Features that Govern Terminator Strength in Plants"). This report describes a high throughput study of Arabidopsis and maize terminators (polyadenylation sites) that entails the use of so-called STARR-Seq to evaluate the efficacy of different 3' regions (terminators). These 3' regions include those from a wide range of

Arabidopsis and maize genes as well as synthetic 3' regions whose design is guided by several principles. The authors find that terminator strength in their assay, which reflects the ability of a terminator to increase overall reporter gene expression, is largely determined by the occurrence of well-established polyadenylation-associated motifs that in turn promote efficient cleavage and polyadenylation within the respective 3' regions. The authors find some interesting differences in trends in terminators from Arabidopsis and maize genes and their efficiencies in tobacco and maize cells. Features that correlate with high terminator efficiency can be used to predict terminator function, and to successfully "evolve" high-efficiency terminators. Taken together, these results confirm a host of "low-throughput" work that spans the past 35 years or so and provide a foundation by which synthetic elements suited for high-level expression may be created.

The authors have addressed the most salient criticisms and returned a much-improved manuscript. To summarize (briefly):

A main fault in the first submission was a very large discrepancy between newly-measured polyadenylation efficiencies for the t35S element with those reported in several other studies. The authors describe and correct an error in their analysis that resolved this issue. They also provide additional experimental support that corroborates their measurements. Thus, this issue has been resolved.

The authors also add results from an orthogonal assay of terminator efficiency, using a dual-luciferase assay to corroborate the values obtained by STARR-Seq. This is an excellent addition and lends much confidence in the conclusions drawn by the authors.

The authors have greatly expanded the context for this study, citing and considering research that is foundational for the current high-throughput study. The correspondence between "historical" studies that focus on single genes/elements and the STARR-Seq results is gratifying and lends credence to the conclusions drawn by the authors.

I have two questions or comments that the author may wish to address:

In looking into agreement with other reports, I notice that the terminator the authors use from the HSP18.2 gene, one reported by several groups as being an exceptionally strong terminator, is not actually the same as that described by others (Felippes et al. 2022 and references therein). The element used in the STARR-Seq study is located well within the CDS of this gene and is not the same strong terminator that is situated in the 3'-UTR. The authors may want to make this distinction somewhere, since the community that is interested in plant terminators may see the extremely low

efficiency - -4.22 (taken from Supplementary Table 2) - for HSP18.2 (AT5G59720) and suspect that the STARR-Seq results may not be reliable.

I was wondering if the authors had further analyzed the “authentic” terminators according to their position along a gene (5'-UTR, CDS, intron, etc.). According to Supplementary Fig. 1C, a sizeable fraction of the Arabidopsis terminators are from annotated CDS and TSS. Wu et al. (2011) noted that CDS-situated poly(A) sites have a distinctive nucleotide composition. Are there discernible differences between sites in the different locations when it comes to terminator or cleavage efficiencies?

Reviewer #1 (Remarks to the Author):

The authors' study delves into the influence of 3' UTR sequences on gene expression, with a focus on their composition, GC content, and motifs, making a significant contribution to our understanding of post-transcriptional regulation. Despite addressing some concerns raised during the peer review, the revised manuscript still harbors critical issues that warrant further attention. My primary concern revolves around the use of STARR-seq to evaluate "Terminator strength." In particular, I question the validity of using RNA-to-DNA ratios as a metric. Although this ratio is a reliable indicator of promoter activity—where a stronger promoter results in higher RNA levels—this logic may not extend to 3' UTR sequences functioning as termination signals. Contrary to expectations, higher RNA production does not necessarily imply more efficient termination. In fact, one could argue that effective termination might lead to reduced transcript levels downstream.

1. Regarding the response to question 2, "Therefore, we use reporter gene expression as a proxy for the efficiency with which a terminator is cleaved and polyadenylated," employing gene expression as a surrogate for terminator efficiency remains a matter of debate. de Felippes and Waterhouse (2023) discussed how poly(A) tail length affects gene expression, and Bernardes and Menossi (2020) examined the role of 3' Regulatory Regions in gene expression. Nevertheless, these studies fall short of providing concrete evidence that transcript levels can reliably estimate terminator efficiency.

2. In response to question 6, "Our study primarily investigates the impact of terminators on mRNA transcript levels, polyadenylation, and cleavage, rather than on transcription termination itself," it appears the essence of the study is to examine the effects of 3' UTR sequences on gene expression, not to directly assess their termination efficiency. It would be beneficial to make this distinction clearer in the manuscript to ensure readers fully understand the innovative approach and its significance for elucidating the role of 3' UTRs in gene regulation.

RESPONSE:

We thank the reviewer for their kind assessment that our manuscript makes important contributions to our understanding of post-transcriptional regulation.

The reviewer correctly states that our study examines the effects of 3'UTR sequences on gene expression, in addition to also capturing their effect on RNA stability and cleavage probability. We also conducted a dual-luciferase assay in both tobacco leaves and maize protoplasts for several weak, intermediate, and strong terminators, and observed strong correlation between their terminator strength as defined below and protein levels measured as nanoluciferase activity (Supplementary Fig. 3a, c).

To avoid the confusion that the reviewer rightly notes, we do not use the term "terminator efficiency" anywhere in the manuscript.

In the manuscript, we clearly define our usage of the term “terminator strength” on **page 3** and **lines 133-137**:

“We define terminator strength as the enrichment of barcodes in RNA over DNA normalized to the enrichment of a control construct containing the 35S terminator. Thus, terminator strength in our assay reflects both transcriptional activity and RNA stability, and tends to have low values due to the normalization to the highly active 35S terminator”.

This definition is repeated in the legend for Figure 1 and in the methods section. As requested by the reviewer, we have added the following text to the discussion to further clarify our meaning.

Page 11, lines 485-487:

“Here, we adapted Plant STARR-seq to characterize and optimize plant terminator sequences. Our results capture the effects of these sequences on RNA expression, stability and cleavage probability.”

Reviewer #3 (Remarks to the Author):

This manuscript is a revision of an earlier one (previously entitled “Features that Govern Terminator Strength in Plants”). This report describes a high throughput study of Arabidopsis and maize terminators (polyadenylation sites) that entails the use of so-called STARR-Seq to evaluate the efficacy of different 3’ regions (terminators). These 3’ regions include those from a wide range of Arabidopsis and maize genes as well as synthetic 3’ regions whose design is guided by several principles. The authors find that terminator strength in their assay, which reflects the ability of a terminator to increase overall reporter gene expression, is largely determined by the occurrence of well-established polyadenylation-associated motifs that in turn promote efficient cleavage and polyadenylation within the respective 3’ regions. The authors find some interesting differences in trends in terminators from Arabidopsis and maize genes and their efficiencies in tobacco and maize cells. Features that correlate with high terminator efficiency can be used to predict terminator function, and to successfully “evolve” high-efficiency terminators. Taken together, these results confirm a host of “low-throughput” work that spans the past 35 years or so and provide a foundation by which synthetic elements suited for high-level expression may be created.

The authors have addressed the most salient criticisms and returned a much-improved manuscript. To summarize (briefly):

A main fault in the first submission was a very large discrepancy between newly-measured polyadenylation efficiencies for the t35S element with those reported in several other studies. The authors describe and correct an error in their analysis that resolved this issue. They also provide additional experimental support that corroborates their measurements. Thus, this issue has been resolved.

The authors also add results from an orthogonal assay of terminator efficiency, using a dual-luciferase assay to corroborate the values obtained by STARR-Seq. This is an excellent addition and lends much confidence in the conclusions drawn by the authors.

The authors have greatly expanded the context for this study, citing and considering research that is foundational for the current high-throughput study. The correspondence between “historical” studies that focus on single genes/elements and the STARR-Seq results is gratifying and lends credence to the conclusions drawn by the authors.

I have two questions or comments that the author may wish to address:

In looking into agreement with other reports, I notice that the terminator the authors use from the HSP18.2 gene, one reported by several groups as being an exceptionally strong terminator, is not actually the same as that described by others (Felippes et al. 2022 and references therein). The element used in the STARR-Seq study is located well within the CDS of this gene and is not the same strong terminator that is situated in the 3'-UTR. The authors may want to make this distinction somewhere, since the community that is interested in plant terminators may see the extremely low efficiency - -4.22 (taken from Supplementary Table 2) - for HSP18.2 (AT5G59720) and suspect that the STARR-Seq results may not be reliable.

I was wondering if the authors had further analyzed the “authentic” terminators according to their position along a gene (5'-UTR, CDS, intron, etc.). According to Supplementary Fig. 1C, a sizeable fraction of the *Arabidopsis* terminators are from annotated CDS and TSS. Wu et al. (2011) noted that CDS-situated poly(A) sites have a distinctive nucleotide composition. Are there discernible differences between sites in the different locations when it comes to terminator or cleavage efficiencies?

RESPONSE:

We thank the reviewer for their kind assessment of the revised manuscript, and we thank them again for their careful review and suggestions for improvement.

Indeed, the reviewer spotted another result in our data that needs clarification. As we describe in the methods, the tested *Arabidopsis* sequences are largely derived from experimental data (Thomas et al. 2012). We can only speculate why this study did not detect the canonical HSP18.2 (AT5G59720) cleavage site; for example, HSP18.2 might be too weakly expressed in the conditions used. We now note in the methods that we have not applied additional filtering steps to this data, and we point to the tested HSP18.2 sequence as an example of an included coding sequence.

Page 12, Lines 553-563:

“For *Arabidopsis*, we used experimentally determined cleavage and polyadenylation sites described by Thomas *et al.* (Thomas *et al.* 2012). We selected the primary cleavage and polyadenylation sites for each gene ($n = 18,450$) in wild-type plants, as well as prominent secondary cleavage sites that had at least 30% of total reads per gene ($n = 2,325$). For the 3,754 genes without an experimentally defined cleavage site, we utilized the end of their 3' UTR annotation in the *Arabidopsis* TAIR10 annotation as the cleavage site. This yielded a total of 24,529 *Arabidopsis* terminator sequences (Supplementary Table 1). No additional filtering was applied, leading to the inclusion of some coding sequences. One example is the tested

HSP18.2 sequence, which differs from the strong terminator sequence described in prior studies (Felippes 2022).”

To address their second point: first, the original Supplementary Figure 1C subtly misrepresents the fraction of Arabidopsis cleavage sites (as stated in the figure legend) residing in annotated CDS and TSS regions. The original pie chart shows that 12% of the total bases in the library reside in the CDS rather than 12% of terminators residing in the CDS. Because Arabidopsis is such a gene-rich, dense genome, some of the 170 bp terminator sequences will overlap with coding sequence, even though their respective cleavage site resides in the 3'UTR as expected. We have corrected this pie chart so that the figure reflects the provided figure legend. Only ~5% of cleavage sites reside in coding sequence (see below Supplementary Figure panel 1c).

Distribution of cleavage sites in Arabidopsis dataset (n=24,529)

Second, as suggested by the reviewer, we have compared the terminator sequences from the Thomas dataset according to their position along the gene and included our control coding sequences in the analysis. As expected, coding sequences are weak terminators (**Response Figure 1A**), even though they are somewhat stronger than control CDS regions (**Response Figure 1B**). The cleavage probability of coding sequences is also weaker than for any other sequence type (**Response Figure 1C**). However, the cleavage probability of these sequences is significantly higher than that of control CDS regions (**Response Figure 1D**), consistent with how they were identified by Thomas et al.

Response Figure 1. Sequences with annotated cleavage sites in CDS are poor terminators but differ from control coding regions in cleavage probability. A) Terminator strength and C) cleavage probability (in tobacco) for *Thomas et al 2012 Arabidopsis* terminator sequences. B) Terminator strength and D) cleavage probability of *Arabidopsis* terminator sequences annotated in the CDS (coding) vs control CDS terminators.

We have also examined the nucleotide composition of the experimentally derived CDS “terminators”. They show a distinct nucleotide composition compared to control CDS regions (**Response Figure 2**, but a similar nucleotide composition compared to terminators overall, see **Response Figure 3**, *Thomas et al. 2012* dataset)

Response Figure 2. Terminators from *Thomas et al 2012* that are annotated inside CDSs are distinct from random CDS controls. A) The nucleotide frequency plot of terminators tested in library in which the cleavage site is annotated inside a CDS region. B) The nucleotide frequency of terminators tested in library that were derived from CDS regions as controls.

Response Figure 3. Terminator sequences derived from *Thomas et al 2012* resemble canonical patterns of cleavage and polyadenylation sites better than terminators based on 3'UTR annotation in ARAPORT11 and TAIR10.

References:

Loke, Johnny C., Eric A. Stahlberg, David G. Strenski, Brian J. Haas, Paul Chris Wood, and Qingshun Quinn Li. 2005. "Compilation of MRNA Polyadenylation Signals in Arabidopsis

Revealed a New Signal Element and Potential Secondary Structures.” *Plant Physiology* 138 (3): 1457–68.

Thomas, Patrick E., Xiaohui Wu, Man Liu, Bobby Gaffney, Guoli Ji, Qingshun Q. Li, and Arthur G. Hunt. 2012. “Genome-Wide Control of Polyadenylation Site Choice by CPSF30 in *Arabidopsis*.” *The Plant Cell* 24 (11): 4376–88.

REVIEWERS' COMMENTS

Reviewer #1 (Remarks to the Author):

The authors have addressed all of my comments.

Reviewer #3 (Remarks to the Author):

The authors have addressed all of my concerns and questions. I believe the revised manuscript is an interesting addition to the body of literature pertaining to mRNA polyadenylation in plants.

NCOMMS-23-26889 RESPONSE TO REFEREES

Reviewer #1 (Remarks to the Author):

The authors have addressed all of my comments.

We thank you for the insightful review of our work and the close attention to detail. We are eternally grateful for your review.

Reviewer #3 (Remarks to the Author):

The authors have addressed all of my concerns and questions. I believe the revised manuscript is an interesting addition to the body of literature pertaining to mRNA polyadenylation in plants.

We thank you for your kind words and for your careful assessment of our work. Peer review here made our paper stronger and we owe it all to our reviewers and editors.